# Interferon lambda 4 impairs hepatitis C viral antigen presentation and attenuates T cell responses

Qian Chen[1], Mairene Coto-Llerena [1], Aleksei Suslov[1], Raphael Dias Teixeira [2], Isabel Fofana[1], Sandro Nuciforo [1], Maike Hofmann [3], Robert Thimme [3], Nina Hensel [3], Volker Lohmann [4], Charlotte K. Y. Ng [5], George Rosenberger[1], Stefan Wieland [1] & Markus H. Heim [1,6✉]

Genetic variants of the interferon lambda (IFNL) gene locus are strongly associated with spontaneous and IFN treatment-induced clearance of hepatitis C virus (HCV) infections. Individuals with the ancestral IFNL4-dG allele are not able to clear HCV in the acute phase and have more than a 90% probability to develop chronic hepatitis C (CHC). Paradoxically, the IFNL4-dG allele encodes a fully functional IFNλ4 protein with antiviral activity against HCV. Here we describe an effect of IFNλ4 on HCV antigen presentation. Only minor amounts of IFNλ4 are secreted, because the protein is largely retained in the endoplasmic reticulum (ER) where it induces ER stress. Stressed cells are significantly weaker activators of HCV specific CD8[+] T cells than unstressed cells. This is not due to reduced MHC I surface presentation or extracellular IFNλ4 effects, since T cell responses are restored by exogenous loading of MHC with HCV antigens. Rather, IFNλ4 induced ER stress impairs HCV antigen processing and/or loading onto the MHC I complex. Our results provide a potential explanation for the IFNλ4–HCV paradox.

[1] Department of Biomedicine, University of Basel, Basel, Switzerland. [2] Biozentrum, University of Basel, Basel, Switzerland. [3] Department of Medicine II, University Hospital Freiburg, Freiburg, Germany. [4] Department of Infectious Diseases, Molecular Virology, Centre for Integrative Infectious Disease Research (CIID), University of Heidelberg, Heidelberg, Germany. [5] Department for BioMedical Research (DBMR), Oncogenomics Lab, University of Bern, Bern, Switzerland. [6] Clarunis, University Center for Gastrointestinal and Liver Diseases, Basel, Switzerland. ✉email: markus.heim@unibas.ch

nterferon lambda (IFNλ) is generally recognized for its important role in innate immunity at mucosal barriers of the intestine and the respiratory tract[1], but it is also an important regulator of the host response to Hepatitis C virus (HCV) infection. In 2009, genome-wide association studies unexpectedly linked genetic variants of the IFN-lambda (IFNL) gene locus with spontaneous and treatment-induced clearance of HCV[2–7]. Four years later, IFNλ4 was discovered and identified as the molecular link between genotype and phenotype[8]. IFNλ4 differs in several important ways from the other three IFNλ family members. IFNλ4 shares only 29% amino acid identity with its closest relative IFNλ3, whereas the three other members of the family share >80%[8]. Compared with the other IFNλ family members, IFNλ4 is very weakly induced by viral infections, poorly expressed, and hardly secreted at all[9–14]. These properties of IFNλ4 make it difficult to explore functional properties that could explain its negative impact on defense against HCV. Years after its discovery, the most compelling evidence for the central role of IFNλ4 still comes from genetic association studies. Genetic variations in the IFNL4 gene encode several isoforms. The IFNL4 signature single nucleotide polymorphism (SNP) at rs368234815 is an insertion/substitution mutation (IFNL4-G/TT) that disrupts the open reading frame of the ancestral IFNL4-G allele in exon 1. A second functionally important SNP at rs117648444 (T/A) changes a proline at amino acid position 70 of the protein to serine, resulting in a much less active IFNλ4-S70[15]. Paradoxically, the non-functional IFNL4-TT, the functionally impaired IFNL4-S70, and the fully functional IFNL4-P70 alleles are strongly associated with high, intermediate, and low spontaneous HCV clearance rates, respectively[15]. Why a functional IFN with strong in vitro antiviral activity[10–12] can be a disadvantage in the host response to HCV remains unexplained.

In the present study, we use biochemical and immunological approaches to explore the biological properties of IFNλ4. We confirm previous reports showing that IFNλ4 is poorly secreted, but once secreted, binds to the IFNλ receptor, strongly induces Jak-STAT signaling, and has strong antiviral activity. Here we describe that these positive effects on innate immunity are counteracted by a negative impact on cellular immunity. This negative effect comes from the non-secreted IFNλ4 that accumulates in the endoplasmic reticulum (ER) and induces ER stress. ER stress leads to reduced antigen processing from endogenous HCV proteins, which in turn inhibits HCV antigen presentation to CD8$^+$ T cells. As a consequence, HCV-specific T cell responses are attenuated. Our findings provide mechanistic evidence that IFNλ4 has a negative impact on antigen-dependent immune responses that could explain the "IFNλ4 paradox".

## Results

### IFNλ4 is poorly secreted but stimulates potent JAK-STAT signaling.
The first description of IFNλ4 in 2013 reported that transfection of cells with an IFNλ4 expression plasmid induced the phosphorylation of the classical IFN signal transducers STAT1 and STAT2 despite the fact that IFNλ4 protein could not be detected in the supernatant of the transfected cells[8]. In order to obtain further insights into IFNλ4 production, glycosylation, and secretion, we transiently transfected Huh-7 cells with expression vectors encoding different Myc-tagged IFNs and monitored intracellular and extracellular accumulation. Forty-eight hours (h) post-transfection, total cell lysates and cell-culture supernatant were subjected to Myc-tag-specific (Fig. 1a, top panel) and IFN-specific (Fig. 1a, bottom panel) western blot analysis. As expected, IFNα and IFNλ1, 2, and 3 accumulated in the cell-culture supernatant (Fig. 1a). In contrast, IFNλ4 accumulated intracellularly and was barely detectable in the supernatant (Fig. 1a).

The poor secretion of IFNλ4 was not caused by the Myc-tag, because the same results were obtained using Flag-tagged versions of IFNα, IFNλ1–4 (Supplementary Fig. 1). Next, we investigated the kinetics of IFNλ4 secretion. To investigate the IFN production rate, we measured IFNλ3 and IFNλ4 production and intracellular accumulation in the cells during 120 h after transfection, and we quantified daily IFN secretion over a 5-day period (Fig. 1b–e). During the entire time of the experiment, IFNλ4 was barely detectable in the supernatant but instead accumulated intracellularly, whereas IFNλ3 was secreted as expected (Fig. 1b–e).

Next, we assessed the activity of the secreted IFNλ4. To that end, we stimulated Huh-7 cells stably expressing the receptor IFNλR1 (Huh7-LR cells[16]) with serial dilutions of supernatant collected from cells transfected with IFNλ4 or IFNλ3 expression plasmids. IFNλ4 samples strongly activated STAT1 and STAT2 phosphorylation (pY-STAT1, pY-STAT2) even in dilutions where the Myc-tagged protein was not detectable (Fig. 2a–c). This experiment was also performed in the presence of B18R, a type I IFN inhibitor[17–19]. Treatment of Huh7-LR cells with B18R prevented IFNα- but not IFNλ3- and IFNλ4-mediated JAK-STAT activation (Fig. 2b, c and Supplementary Fig. 2). To further study the activity of IFNλ4 in a time-dependent manner, we investigated the kinetics of STAT phosphorylation triggered by IFNλ4 or IFNλ3. Compared to IFNλ3, IFNλ4 activated STAT1 and STAT2 faster and stronger (Fig. 2d, e). For a more quantitative assessment of the specific activity of IFNλ3 and IFNλ4, we purified the proteins from cell-culture supernatants (Supplementary Fig. 3). Huh7-LR cells were then stimulated with different concentrations of the purified proteins (Supplementary Fig. 4a, b). Dose-response curves were generated and used to calculate EC50 values. The EC50 for STAT1 phosphorylation was 1.4 and 39.1 ng/ml for IFNλ4 and IFNλ3, respectively, and for STAT2 activation it was 1.9 and 54.3 ng/ml for IFNλ4 and IFNλ3, respectively. Thus, IFNλ4 was approximately 28-fold more potent than IFNλ3. The high specific activity of IFNλ4 was confirmed in Mx1-promoter-reporter gene assays (Supplementary Fig. 4c), by quantifying interferon-stimulated gene (ISG) expression (Supplementary Fig. 4d), and also demonstrated as antiviral activity in HCV replicon cells (Supplementary Fig. 4e). We conclude that IFNλ4 is poorly secreted, but highly potent.

### IFNλ4 glycosylation increases its function.
Next, we wanted to investigate what renders IFNλ4 so potent. It has been described that posttranslational modification of IFNλ4 by N-linked glycosylation is required for secretion but apparently does not influence its activity[10]. Nevertheless, we explored whether glycosylation is also important for the activity of IFNλ4. First, we compared pY-STAT1 activation in Huh7-LR cells that were treated with equal amounts of purified IFNλ4 and IFNλ3 produced in Huh7 cells (i.e., properly glycosylated) or Escherichia coli (E. coli) (i.e., lacking all glycosylation) as quantified by silver staining (Fig. 3a). In case of IFNλ3, which is physiologically not glycosylated[20], there was no difference between E. coli and Huh7-derived purified proteins in terms of STAT1 activation (Fig. 3b). This was different for IFNλ4, where Huh7-derived preparations were much more potent (Fig. 3b), suggesting that glycosylation increased IFNλ4 activity. To further investigate the role of glycosylation for IFNλ4 activity, we pre-treated Huh-7-produced purified IFNλ4 with different glycosidases prior to stimulation of Huh7-LR cells. Consistent with previous reports[10], we observed in western blots a shift of IFNλ4-specific bands to lower molecular weights in samples that were pre-treated with a mix of deglycosylation enzymes (PNGase F, O-glycosidase, neuraminidase, β−1−4 galactosidase, and β-N-acetyl glucosaminidase) or with PNGase F alone (Fig. 3c, lane 2 and 3 from the left),

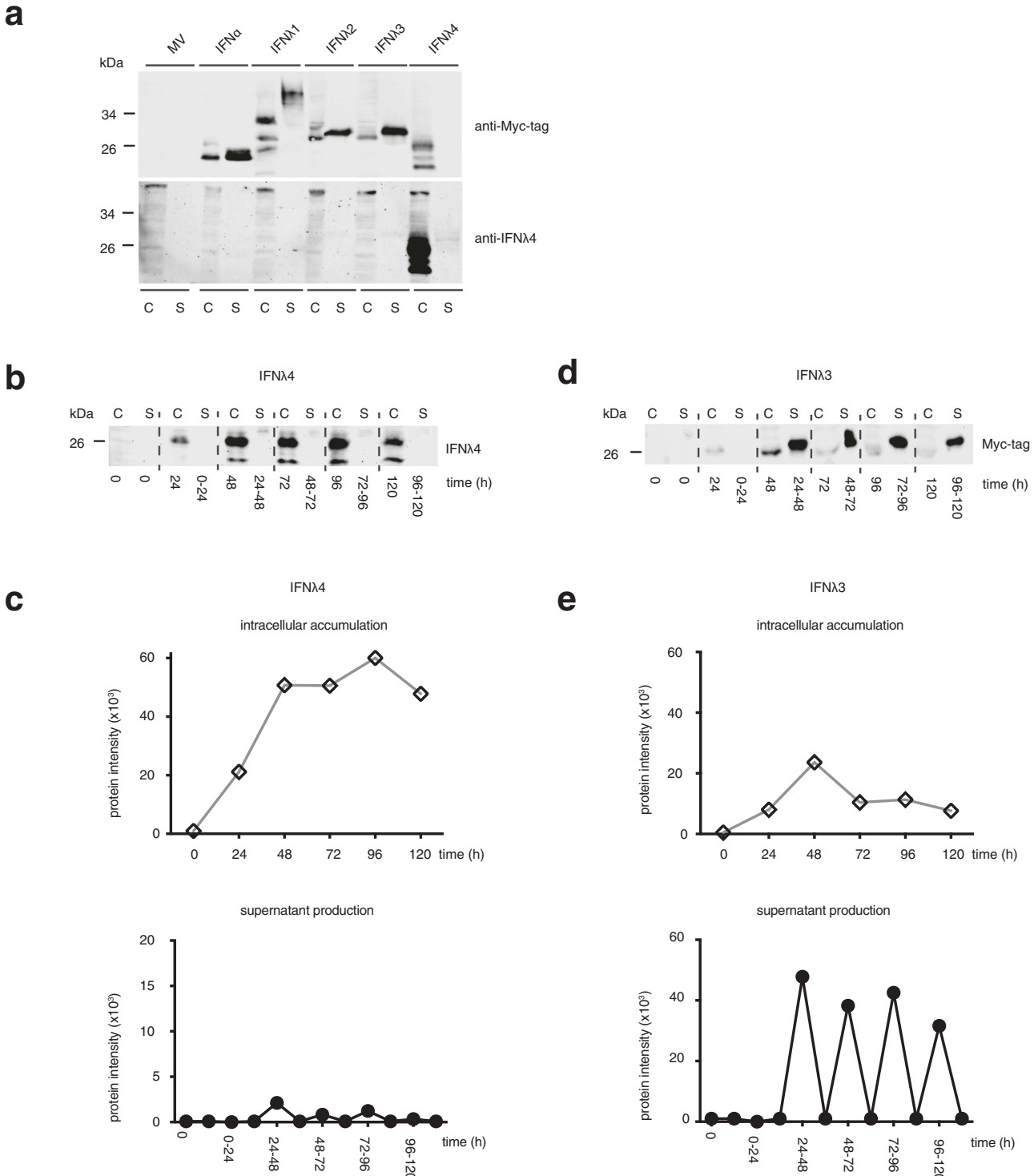

**Fig. 1 IFNλ4 accumulates intracellularly and is poorly secreted compared to other IFNs.** Huh7 cells were mock-transfected (MV) or transfected with Myc-His-tagged expression vectors for IFNα and IFNλ1–4. **a** 48 h post-transfection, cell lysates (C) and supernatants (S) were collected. IFN proteins were detected by Myc-tag specific western blotting with the signal acquisition in the green channel of the Odyssey-CLx image system (Licor) (top panel). Without stripping, membranes were re-probed with an IFNλ4-specific antibody with the signal acquisition in the red channel (bottom panel). In lysates from IFNλ1 and IFNλ4-expressing cells, several bands were detected representing different glycosylation forms of the proteins. Data are representative of $n$ = 3 independent experiments. **b**, **c** IFNλ4 and (**d**, **e**) IFNλ3 total intracellular protein accumulation and daily supernatant production were measured over a 5-day period after transfection. Transfected cell lysates were harvested at the indicated time points after transfection. Cell-culture supernatants were collected every 24 h after a fresh medium was added to the transfected cells. IFNλ4 and IFNλ3 proteins were analyzed in cell lysates (C) and supernatants (S) by (**b**) IFNλ4- and (**d**) Myc-tag-specific western blotting, respectively. kDa, kilodalton. Relative accumulation and daily production of (**c**) IFNλ4 and (**e**) IFNλ3 were calculated from the band intensities in the western blots. Empty diamond, intracellular accumulation of IFNλ4 and IFNλ3. Filled circle, daily production of IFNλ4 and IFNλ3. Data are representative of $n$ = 3 independent experiments. Source data are provided as a Source Data file.

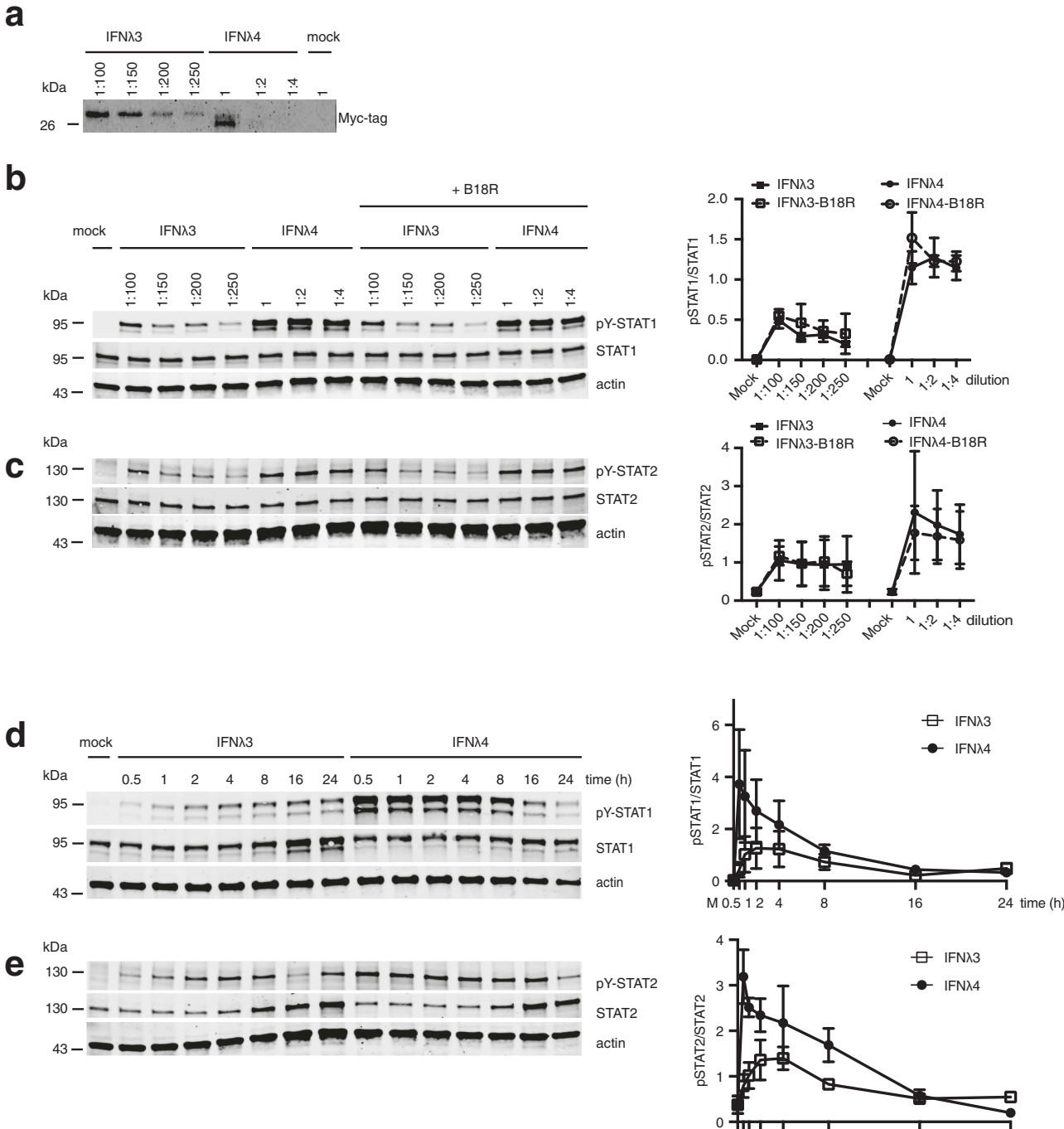

**Fig. 2 IFNλ4 is poorly secreted but very potently activates JAK-STAT signaling. a** Huh7 cells were transfected with IFNλ3-Myc-His and IFNλ4-Myc-His plasmids or mock vector (mock) and cell-culture supernatants were collected 48 h later. IFNλ3 and IFNλ4 containing supernatants were diluted as indicated and 50 μl each was analyzed by Myc-tag-specific western blotting. Data are representative of *n* = 3 independent experiments. **b, c** Huh7-LR cells were seeded in a 48-well plate and pre-treated with B18R (500 ng/ml) for 2 h or left untreated. After washing, cells were stimulated for 30 min with 150 μl of IFNλ3 and IFNλ4 containing supernatants at the same dilutions shown in (**a**). Stimulated cells were lysed and analyzed for STAT activation by western blotting for (**b**) pY-STAT1 and STAT1 and (**c**) pY-STAT2 and STAT2 and actin as a loading control. The western blot signals were quantified and the ratios of phospho-STAT1/2 to total STAT1/2 were plotted. Results are presented as mean ± SEM of *n* = 3 independent experiments. Solid line with filled squares, IFNλ3; dashed line with empty squares, IFNλ3-B18R; solid line with filled circles, IFNλ4; dashed line with empty circles, IFNλ4-B18R. **d, e** Huh7-LR cells were stimulated with 150 μl of 1:100 diluted IFNλ3 or undiluted IFNλ4 containing supernatant for the indicated time or were mock-treated. STAT1 (**d**) and STAT2 (**e**) activation was assessed as described in (**b, c**). The western blot signals were quantified and the ratios of phospho-STAT1/2 to total STAT1/2 were plotted. Results are presented as mean ± SEM of *n* = 3 independent experiments. Solid line with empty squares, IFNλ3; solid line with filled circles, IFNλ4. Source data are provided as a Source Data file.

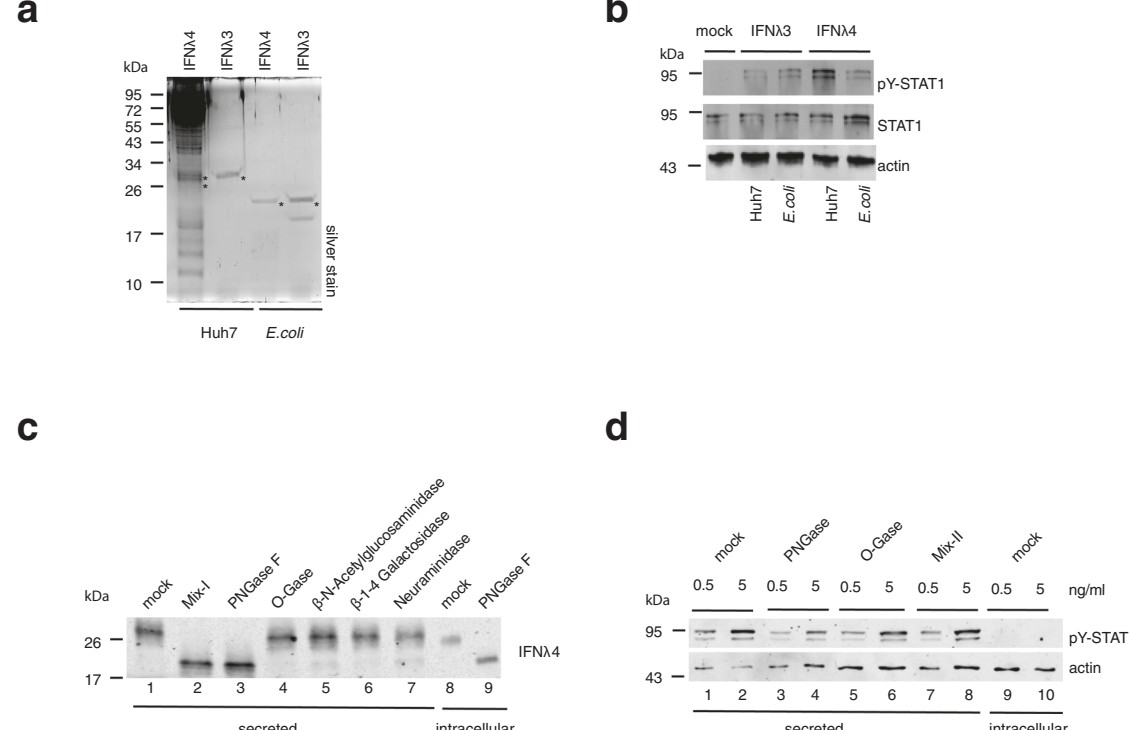

**Fig. 3 IFNλ4 activity is dependent on N-linked glycosylation. a** 50 ng each of IFNλ4 and IFNλ3 purified from Huh7 cells or *E. coli* were analyzed by 12% SDS-PAGE and quantified by silver staining. IFNλ4 and IFNλ3 proteins were indicated by an asterisk (*). Huh7-produced IFNλ3 and IFNλ4 proteins are C-terminally tagged with Myc and 6× His, while *E. coli*-produced IFNλ3 and IFNλ4 proteins are N terminally tagged with 6× His. M, protein molecular weight marker; kDa, kilodalton. Data are representative of *n* = 3 independent experiments. **b** Huh7-LR cells were stimulated for 30 min with 8 ng/ml of IFNλ4 or IFNλ3 purified from Huh7 cells or *E. coli* or with supernatants collected 48 h-post mock transfection (mock) in Huh7 cells. Cells were lysed and analyzed for STAT activation by western blotting for pY–STAT1, STAT1, and actin as a loading control. **c** 10 µl of purified IFNλ4 from Huh7 cell-culture supernatant (secreted) was incubated alone in GlycoBuffer (mock) or was digested with a protein deglycosylation mix (Mix-I, an enzyme mix of PNGase F, *O*-glycosidase, β-*N*-acetyl glucosaminidase, β1–4 galactosidase or α2-3,6,8 neuraminidase), N-linked glycosidase (PNGase F), O-link glycosidase (O-Gase), β-*N*-acetylglucosaminidase, β-1–4 galactosidase or neuraminidase at for 1 h 37 °C. A cell lysate of IFNλ4-transfected-Huh7 cells was incubated in GlycoBuffer (mock) or digested with PNGase F for 1 h at 37 °C. IFNλ4 proteins in the digested samples were detected by IFNλ4-specific western blotting. kDa, kilodalton. **d** Purified IFNλ4 (secreted) and a cell lysate of IFNλ4-transfected-Huh7 cells (intracellular) were mock-treated in GlycoBuffer or treated exactly as described in (**c**) with the indicated glycosidases (PNGase F, O-Gase and Mix-II, an enzyme mix of β-*N*-acetylglucosaminidase, β-1–4 galactosidase, neuraminidase). Huh7-LR cells were stimulated for 30 min at 37 °C with the mock-treated or glycosidase-digested IFNλ4 preparations at 0.5 and 5 ng/ml. Stimulated cells were processed and analyzed as described in (**b**). Data are representative of *n* = 3 independent experiments in (**b–d**). Source data are provided as a Source Data file.

indicating removal of N-linked glycosylation. There was no clear change in molecular weight of IFNλ4 upon treatment with other enzymes (Fig. 3c, lanes 4–7). Of note, PNGase F treatment of intracellular IFNλ4 also shifted it to the same lower band, indicating that intracellular IFNλ4 was also *N*-glycosylated (Fig. 3c, lane 9).

To establish whether glycosylation affects IFNλ4 activity, enzymatically pre-treated IFNλ4 samples were used to stimulate Huh7-LR cells. PNGase F, but not the other enzymes, reduced the activity of IFNλ4, indicating that N-linked glycosylation is important for IFNλ4 activity (Fig. 3d). Of note, intracellular IFNλ4 did not induce pY-STAT1 activation despite the fact that it was *N*-glycosylated (Fig. 3c, d).

**IFNλ4 accumulates in the ER**. The observation that intracellularly accumulated IFNλ4 is *N*-glycosylated refutes the hypothesis that a defect in *N*-glycosylation is responsible for the poor secretion of IFNλ4. To explore other potential mechanisms for intracellular retention of IFNλ4, we first set out to identify the subcellular compartment(s) where IFNλ4 retention occurred. Immunofluorescence studies using IFNλ4 antibodies together with an ER staining dye or an anti-Golgi antibody revealed that

IFNλ4 colocalized both with ER membranes and the Golgi apparatus (Fig. 4a). Colocalization analysis showed preferential accumulation of IFNλ4 in the ER. Next, we performed subcellular fractionation with IFNλ4-transfected and IFNλ3-transfected Huh7 cells. More than 50% of total IFNλ4 was enriched in the ER or ER-PM (plasma membrane) fractions, whereas IFNλ3 was enriched in the cytosolic (Cyto) fraction (Fig. 4b). Of note, intracellular IFNλ3 was as potent as secreted IFNλ3 (Supplementary Fig. 5a, left panel). In contrast, intracellular IFNλ4 was at least 100 times weaker than secreted IFNλ4 irrespective of its subcellular location (Supplementary Fig. 5a, right panel). Together with the results from Fig. 4a, this strongly suggests that intracellularly retained IFNλ4 is not properly folded and remains inactive.

**IFNλ4 induces ER stress**. The accumulation of unfolded/misfolded proteins in the lumen of the ER induces the unfolded protein response (UPR) through the activation of the ER stress sensors protein kinase RNA-like endoplasmic reticulum kinase (PERK), inositol-requiring transmembrane kinase/endoribonuclease 1 (IRE1), and activating transcription factor-6 (ATF6) and their signaling cascades[21,22]. Therefore, we investigated whether these

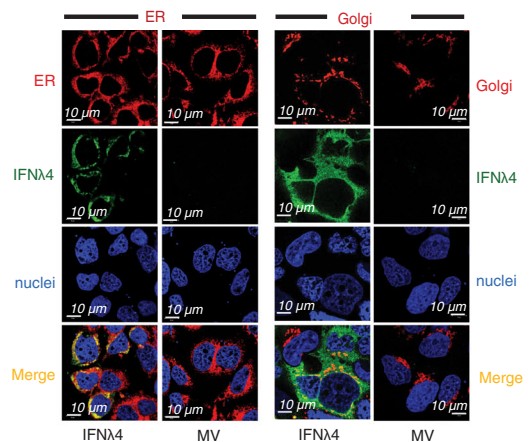

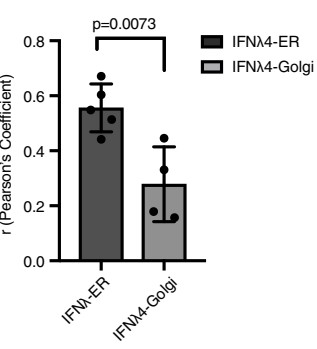

**a**

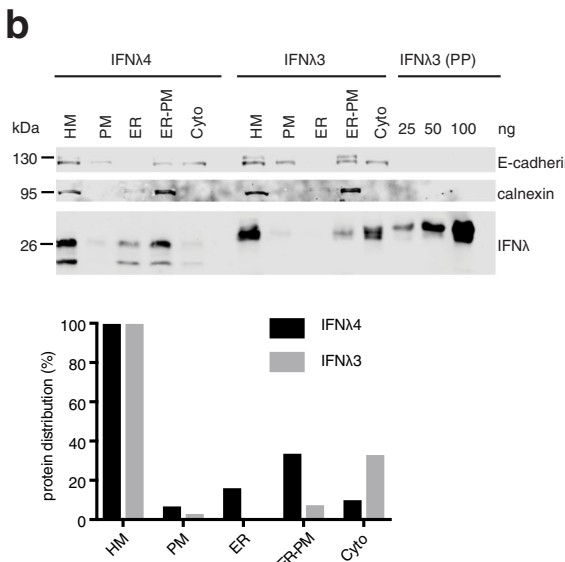

**b**

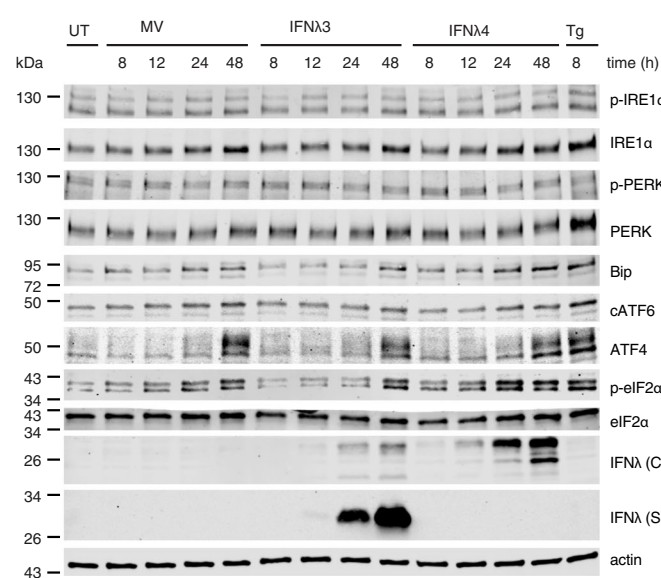

**c**

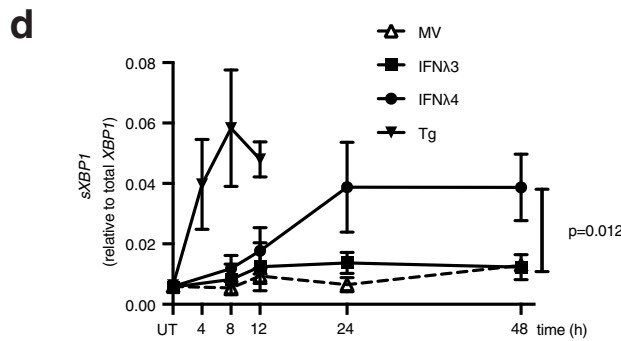

**d**

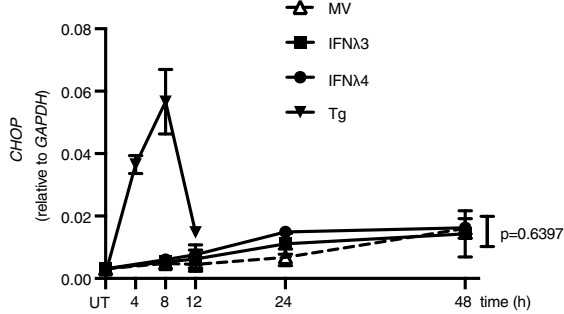

pathways were activated in cells transfected with IFNλ4 expression plasmids. PERK, IRE1, ATF6, and eIF2α-activating transcription factor-4 (ATF4) were not significantly activated upon the expression of IFNλ4 in Huh7 cells. Of note, treatment of Huh7 cells with the ER stress inducer Thapsigargin did also not affect these proteins. We, therefore, used phosphorylation of eIF2α (p-eIF2α), splicing of

X box-binding protein (XBP1), and CHOP mRNA expression as indicators of ER stress induction in Huh7 cells. Transfection of IFNλ4, but not IFNλ3 or other IFN proteins, induced transient phosphorylation of eIF2α with a peak at time point 24 h (Fig. 4c and Supplementary Fig. 5b) and splicing of XBP1, but did not induce CHOP (Fig. 4d and Supplementary Fig. 5c). Of note,

**Fig. 4 IFNλ4 accumulates in the endoplasmic reticulum (ER) and causes ER stress. a** Huh7 cells were transfected with IFNλ4 encoding or mock expression vectors (MV) and processed 24 h later for immunofluorescence staining of ER (red, Cytopainter ER staining kit), Golgi (red, anti-Giantin antibody), IFNλ4 (green, anti-IFNλ4 antibody), and nuclei (blue, DAPI). Images are representative of $n = 5$ independent experiments. Scale bars are 10 μm. Pearson's correlation of colocalization coefficients for IFNλ4-ER (dark gray) and IFNλ4-Golgi (light gray) were quantified as described in the "Methods" section separately for each IFNλ4-expressing cell, $p = 0.0073$, two-tailed unpaired $t$-test. **b** Huh7 cells were transfected with IFNλ4 or IFNλ3 expression vectors and homogenized 24 h later. Plasma membrane (PM), ER, plasma-ER associated membrane-associated (ER-PM) and cytosolic (Cyto) fractions were enriched from the cell homogenate as described in the "Methods" section. 2 μl of each fraction and dilutions of purified IFNλ3 as a protein standard were analyzed by western blotting for the presence of IFNλ4 and IFNλ3 (using anti-Myc-tag antibody) and for E-cadherin and calnexin as markers for PM and ER, respectively (upper panel). The IFNλ protein distribution in the fractions was calculated from the western blot data with the IFNλ4 (black) and IFNλ3 (light gray) signal in the total HM set to 100% (lower panel). kDa, kilodalton. **c** Huh7 cells were transfected with Myc-His-tagged IFNλ3 and IFNλ4 expression vectors or a mock vector (MV) or left untreated (UT). Cell lysates and cell-culture supernatants were collected at the indicated time points. Cell lysates were analyzed by western blotting for the ER stress markers IRE1α, phosphor- IRE1α, PERK, phosphor-PERK, Bip, cleaved ATF6 (cATF6), ATF4, phospho-eIF2α, and eIF2α using marker-specific antibodies as well as intracellular IFNs (IFNs(C)) using an anti-Myc-tag antibody. IFN proteins in the supernatants (IFNs(S)) were also detected with an anti-Myc-tag antibody. Cells treated with 2 μM Thapsigargin (Tg) for 8 h served as a positive ER stress control. **d** Total cellular RNA was harvested from Huh7 cells transfected or treated as described in (**c**) at the indicated time points. Spliced mRNA of the ER stress marker sXBP1 (left panel) and total mRNA of CHOP (right panel) were analyzed by RT-qPCR as described in the "Methods" section. sXBP1 levels are shown as relative expression to mRNA level of total XBP1. CHOP mRNA expression was calculated relative to that of the housekeeping gene GAPDH. Results are presented as mean±SEM of $n = 3$ independent experiments. The dashed line with empty triangles, MV; Solid line with filled squares, IFNλ3; solid line with filled circles, IFNλ4; solid line with filled triangles, Tg (thapsigargin). Source data are provided as a Source Data file.

stimulation of cells with supernatant of IFNλ4 did not induce *XBP1* splicing (Supplementary Fig. 5c), suggesting that ER stress is caused by misfolded endogenous IFNλ4 and not by the small amounts of correctly folded and secreted IFNλ4. We also quantified induction of *sXBP1* and *CHOP* mRNA in doxycycline-inducible HepG2 cell lines that stably express GFP-tagged IFNλ4 (IFNλ4-GFP) or its truncated non-functional form IFNλ4 p131-GFP. Indeed, wild-type, but not truncated IFNλ4 induced both the early and late UPR, *sXBP1*, and *CHOP* in HepG2 cells (Supplementary Fig. 5d, e).

We conclude that IFNλ4 is not properly folded, is retained in the ER, and induces ER stress.

**Sendai virus infection induces IFNλ4 expression and ER stress.** It has been previously reported that IFNλ4 is only transiently and weakly expressed upon viral infections or stimulation of cells with polyI:C[8,11–13]. Thus, we wanted to confirm the findings obtained in cells transfected with IFNλ4 expression plasmids and further explore the impact of IFNλ4 on the cellular response to Sendai virus (SeV) infection. We first infected the lung epithelial cell line A549 with SeV, and could detect IFNλ4 expression both at the mRNA and the protein level (Supplementary Fig. 6). Of note, SeV infection-induced glycosylated and non-glycosylated IFNλ4 isoforms confirming our findings in cells transfected with IFNλ4 expression vectors (Supplementary Fig. 6b). The poor secretion of IFNλ4 observed in the transfection system (Figs. 1 and 2) was also confirmed. IFNλ4 was only detectable in cell lysates but not in the supernatants (Supplementary Fig. 6b).

To further study the impact of endogenous-induced IFNλ4 on the host response to SeV, we made use of liver-derived organoids (Supplementary Fig. 7)[23,24]. Three liver organoid cell lines of the IFNλ4-expressing genotype *IFNL4-dG* (B16, U15, U12) and three of the IFNλ4 deficient genotype *IFNL4-TT/TT* (U16, nt5, B13) were infected with SeV. Viral replication, IFNβ, IFNλ1, −2/3 and −4 induction and ISG expression were not different between IFNλ4 producing (*dG*) and non-producing (*TT*) organoids (Fig. 5a). This was not due to impaired or defective IFNλ4 protein production, as demonstrated by the detection of IFNλ4 protein in the membrane fraction of *dG* but not *TT* organoids (Fig. 5b). Furthermore, the apparent molecular weight of the IFNλ4 proteins was similar to that in A549 cells at 12 h, demonstrating the IFNλ4 was correctly glycosylated in organoids of the *dG* genotype.

We next assessed whether SeV-induced endogenous IFNλ4 also induced ER stress. Indeed, *sXBP1* mRNA expression was

upregulated in correlation with IFNλ4 expression in the IFNλ4-expressing (*dG*) but not the non-expressing (*TT*) organoids (Fig. 5c). We conclude that IFNλ4 is induced upon SeV infection in *dG* organoids, induces ER stress, but has no detectable impact on SeV replication or the induction of ISGs.

**Impact of IFNλ4 genotype on global gene expression.** To further study the impact of SeV-induced IFNλ4 on the cellular response to viral infection, we performed RNAseq on liver organoids with a *dG* genotype (B20, nt115, U15) and with a *TT* genotype (nt5, U16, U19). From each organoid, we profiled the uninfected organoids at time point 0 h (pre-infection) and 12 h (mock), and from SeV-infected (SeV) organoids at time point 12 h. The 12 h time point was chosen because we observed the peak of IFNλ4 expression at this time (Fig. 5a). Multidimensional scaling (MDS) of the transcriptomic profiles revealed a clear separation between SeV-infected and uninfected samples (Supplementary Fig. 8a). By contrast, among both the infected and the uninfected samples, no clear separation was observed between the IFNλ4 genotypes (Supplementary Fig. 8a). To identify transcriptomic changes induced by SeV infection, we performed a differential expression analysis between SeV-infected and uninfected (pre-infection and mock) samples. We found that SeV infection-induced changes in the expression levels of 4649 and 3306 genes in the *dG* and the *TT* genotypes, respectively, including a significant enrichment of 2983 genes commonly altered in both genotypes (enrichment factor = 2.55, $p < 2.2e-16$, hypergeometric test, Supplementary Fig. 8b). Indeed, the overall gene expression changes induced by SeV were highly similar in the two IFNλ4 genotypes ($r = 0.93$, Pearson correlation, $p < 2.2e-16$, Supplementary Fig. 8b). This was also reflected at the level of biological pathways as determined by pathway enrichment analysis (Supplementary Fig. 8c; Supplementary Data 1 and 2). Since the molecular analysis revealed a dG genotype-specific induction of ER stress, we specifically interrogated the RNAseq data for differences in regulation of genes associated with response to ER stress. Indeed, gene set enrichment analysis revealed that the pathway "GO-Response_To_Endoplasmic_Reticulums_Stress" was positively enriched in organoids of *dG* genotype, although the difference did not reach statistical significance. Nevertheless, the regulation of the corresponding genes was stronger in the organoids of the *dG* genotype (Supplementary Fig. 8d) consistent with the induction of an IFNλ4-dependent ER stress upon viral infection.

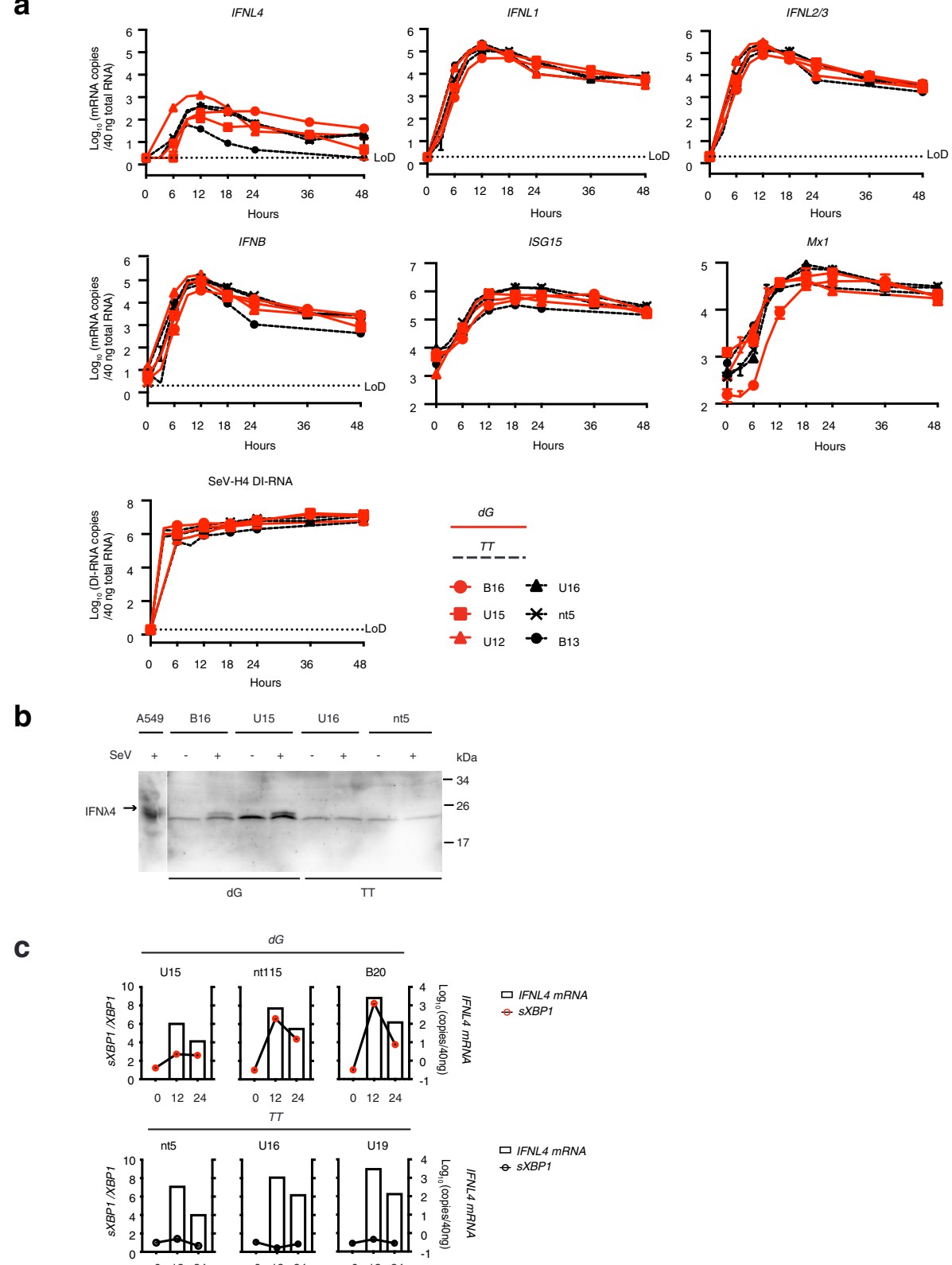

**IFNλ4 expression inhibits HCV peptide-specific CD8+ T cell activation**. Cytosolic antigens presented by MHC-I are primarily derived from proteasomal cleavage of proteins into peptides that are then imported into the ER and loaded onto newly synthesized MHC-I[25]. Thus, we hypothesized that the IFNλ4-induced ER stress could interfere with HCV antigen presentation, and thereby reduce the activation of HCV-specific CD8+ T cells. To address this question, we made use of a unique in vitro cell coculture system consisting of subgenomic HCV replicon cells stably transduced with HLA-A2 (Huh7$_{A2}$HCV$_{EM}$) and an HLA-matched HCV-specific CD8+ T cell clone[26].

**Fig. 5 IFNλ4 is induced upon viral infection in liver-derived organoids.** Liver-derived organoid cultures of the *dG* (B16, U15, U12) and *TT* (U16, nt5, B13) genotype were infected with Sendai virus H4 at a MOI = 10 as described in the "Methods" section. **a** Organoids were harvested at 3, 6, 9, 12, 18, 24, 36, and 48 h after SeV inoculation. Uninfected organoids were harvested at the time of SeV inoculation and used as time point 0 h. Total cellular RNA was extracted and analyzed by RT-qPCR for the expression of SeV DI-RNA and mRNA expression of interferons (*IFNB, IFNL1, IFNL2/3, IFNL4*) and the ISGs *Mx1* and *ISG15*. Results are presented as mean ± SEM of biological replicates (*n* = 3 for B16, U15, U16, and nt5, *n* = 1 for U12 and B13). For better visualization, the data point symbols of the 3 and 9 h time points have been omitted from the graphs. Red, *dG* organoids; black, *TT* organoids; dotted line, lower limit of detection (LoD); negative data points and those below LoD are displayed at LoD. **b** Some SeV-infected organoids (B16, U15, U16, nt5) were harvested at 12 h post-infection and processed for IFNλ4-specific western blot analysis as described in the "Methods" section. M, molecular size marker. Data are presented from one representative experiment. **c** Three organoid cultures each of the *dG* (B20, nt115, U15) and *TT* (nt5, U16, U19) genotype were infected with Sendai virus H4 (MOI = 10). Total cellular RNA was extracted from infected cells at the indicated time points and analyzed for the mRNA expression of *IFNL4* and *sXBP1* mRNA. *sXBP1* levels are shown as the ratio of the spliced *(s)XBP1* mRNA to total *XBP1* RNA. The ratio at the time point 0 of each organoid line was set to 1. *IFNL4* levels were expressed as copy numbers per 40 ng of total RNA. Open bars, *IFNL4* mRNA; lines with red circles, *sXBP1* mRNA in *dG* organoids; lines with black circles, *sXBP1* mRNA in *TT* organoids. Source data are provided as a Source Data file.

We tested the impact of ER stress on T cell response in this system by Thapsigargin treatment of the Huh7$_{A2}$HCV$_{EM}$ (Supplementary Fig. 9a, b). As expected, Thapsigargin induced XBP1 splicing and indeed, HCV-specific CD8$^+$ T cell activation was significantly reduced. Next, Huh7$_{A2}$HCV$_{EM}$ cells were transfected with expression plasmids for IFNα, IFNλ1, IFNλ3, and IFNλ4. Consistent with our previous results, IFNλ4 was retained in the cells and induced ER stress (Supplementary Fig. 10a, b). Consequently, we observed a significant reduction of CD8$^+$ T cell activation by Huh7$_{A2}$HCV$_{EM}$ cells expressing IFNλ4, but not by cells transfected with IFNλ1 or IFNλ3 (Fig. 6a). This was not due to reduced expression of HLA-2A (Fig. 6b), and also not caused by a stronger inhibition of HCV replication in IFNλ4-transfected cells compared to IFNλ1- or IFNλ3-transfected cells (Fig. 6c). Furthermore, treatment of Huh7$_{A2}$HCV$_{EM}$ cells with IFNλ3 or IFNλ4 did not induce an ER stress response in Huh7$_{A2}$HCV$_{EM}$ and had no impact on T cell activation (Supplementary Fig. 9a, b). To further test the hypothesis of IFNλ4 interference with endogenous peptide loading on MHC-I, we investigated whether the T cell response could be rescued by exogenous loading of an HCV peptide to the replicon cells. To this end, we used another replicon cell clone (Huh7$_{A2}$HCV) presenting a mismatched epitope that does not induce T cell activation but is otherwise identical to Huh7$_{A2}$HCV$_{EM}$ cells[26]. As expected, IFNλ4 had no impact on T cell activation in this setting (Fig. 6d). Collectively, these results provide strong evidence that IFNλ4-induced ER stress impairs endogenous processing and MHC-I loading of HCV antigens, and thereby inhibits HCV-specific CD8$^+$ T cell responses.

**IFNλ4-TT expression does not inhibit HCV peptide-specific CD8$^+$ T cell activation.** Finally, we explored the impact of expressing the non-functional TT isoform of IFNλ4 on HCV-specific T cell activation. Cells transfected with an IFNλ4-TT expression plasmid showed no activation of the Jak-STAT pathway and no ER stress response (Fig. 7a, b and Supplementary Fig. 11). Interestingly, this non-functional IFNλ4 is secreted (Fig. 7a). Contrary to IFNλ4, expression of IFNλ4-TT in Huh7$_{A2}$HCV$_{EM}$ cells did not inhibit HCV-specific CD8$^+$ T cell responses (Fig. 7c). These results demonstrate a genotype-specific inhibition of HCV-specific T cell responses that can explain the association of the *IFNL4-dG* genotype with viral persistence.

## Discussion

A number of genetic variants in the IFN-lambda (*IFNL*) gene locus has been associated with the outcome of HCV infection. For example, multiple SNPs in the *IFNL3* have been shown to regulate *IFNL3* transcription or translation[27–31]. In a seminal discovery, IFNλ4 has been identified as the molecular link between the various genetic variants of the IFN-lambda (*IFNL*) gene locus

and spontaneous and treatment-induced clearance of HCV leading to the so-called "IFNλ4 paradox"[8,10,32]. Individuals with a functional IFNL4 gene show an impaired host defense during acute hepatitis C and often become chronically infected[2–7], despite the fact that cell-culture experiments demonstrate a strong antiviral activity of IFNλ4 against HCV and corona virus[10]. Moreover, IFNλ4 signals through the same receptor that other members of type III IFN do, and its biochemical characteristics are highly comparable[10–12,15]. A striking difference to the other IFNλs is the poor secretion of IFNλ4[8,10,11,14]. Therefore, we performed biochemical and immunological studies to explore whether there are differences between IFNλ4 and other IFNs in regard to biosynthesis, secretion, induction of JAK-STAT signaling, and antigen presentation that could explain the negative impact of IFNλ4 on the immune response to the virus.

In our long-term expression experiments over 5 days, IFNλ4 was barely detectable in the cell-culture supernatant at all time points (Fig. 1). Nevertheless, the small amounts of IFNλ4 were highly potent (Fig. 2). Using purified proteins, we estimate that IFNλ4 is 28 times more potent per ng of protein compared to IFNλ3. The reasons for this striking difference in potency remain to be explored. Interestingly, Huh7-produced IFNλ4 that is properly glycosylated is the most potent, while *E. coli*-produced IFNλ4 and IFNλ3 are equally active as Huh7-produced IFNλ3. We extended these observations by comparing IFNλ4 activity pre- and post-enzymatic removal of N-linked sugar residue, demonstrating that glycosylation is important to preserve biological potency of IFNλ4, which has also been demonstrated with other cytokines and receptors[33–35].

Immunofluorescence staining and subcellular fractionation analysis revealed that IFNλ4 is retained in the ER (Fig. 4). It is well known that for some proteins, glycosylation is an important step for entering the secretory pathway. However, the absence of IFNλ4 glycosylation cannot be the main reason for poor secretion, since intracellular retained IFNλ4 seems to be correctly glycosylated (Fig. 3c). Rather, we believe that IFNλ4 is not properly folded. This hypothesis is supported by the observation that even correctly glycosylated but intracellularly retained IFNλ4 cannot induce STAT1 phosphorylation (Fig. 3d). Since proteins have to pass a "quality-control" check before being transported from ER through the protein secretory pathway[36,37], misfolding in the ER could well explain the retention of IFNλ4 in the ER. Like other misfolded glycoproteins, improperly folded IFNλ4 would have to cycle through the "refolding" process before it reaches its native conformation. Only the small fraction of correctly folded protein would then be secreted. The model is consistent with our observation that ER-accumulated IFNλ4 induces UPR in Huh7, HepG2, and HCV replicon (Huh7$_{A2}$HCV$_{EM}$) cells by stimulating *XBP1* mRNA splicing. Many of the ER stress response markers were not strongly induced in the Huh7 cells by IFNλ4, but this was also the case for thapsigargin (Fig. 4c, d). We,

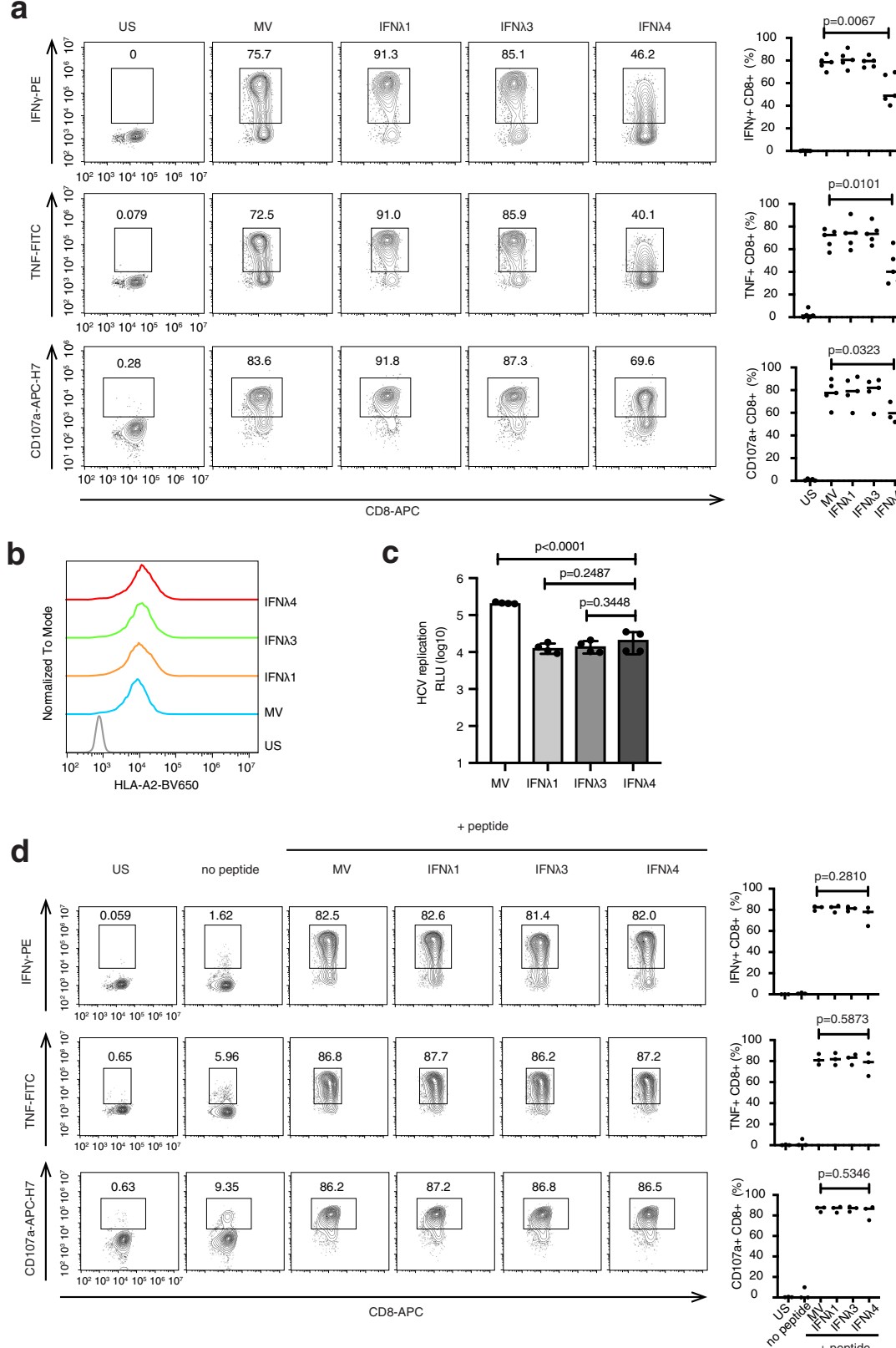

therefore, think that the weak induction of these ER stress response markers rather reflects an intrinsic property of the Huh7 cells than a limited ER stress-inducing activity of IFNλ4. Furthermore, ER stress was not only observed in overexpression systems, but also when IFNλ4 was expressed from the cellular genome in reaction to SeV infection. The fact that *XBP1* mRNA splicing upregulated in viral-

infected organoids of *dG* genotype, but not of *TT* genotype (Fig. 5c), proves that IFNλ4 protein expression is causing ER stress.

SeV-induced IFNλ4 expression in organoids also allowed us to explore the impact of IFNλ4 on global gene expression in a physiological context. As expected, multidimensional scaling (MDS) revealed a clear separation between SeV-infected and

**Fig. 6 IFNλ4 inhibits HCV peptide-specific CD8$^+$ T cell activation. a** Huh7$_{A2}$HCV$_{EM}$ cells were transfected with expression vectors encoding IFNλ1, IFNλ3, IFNλ4 or a mock vector (MV) and 48 h later were cocultured with a NS5B$_{2594-2602}$-specific CD8$^+$ T cell clone for 5 h. Intracellular IFNγ and TNF production, and CD107a mobilization were then analyzed by flow cytometry as described in the "Methods" section. US, unstimulated CD8$^+$ T cell control. IFNλ4 expression inhibits CD8$^+$ T cell activation as measured by the frequency of IFNγ-, TNF- and CD107a-positive CD8$^+$ T cells (left panel). Data of $n = 5$ independent experiments were pooled and are presented as mean ± SEM (right panel). IFNγ$^+$CD8$^+$: $p = 0.0067$, MV vs IFNλ4; TNF$^+$CD8$^+$: $p = 0.0101$, MV vs IFNλ4; CD107a$^+$CD8$^+$: $p = 0.0323$, MV vs IFNλ4, all by two-tailed unpaired $t$-test. **b** Surface expression of HLA-A2 is not affected by IFN expression in vector-transfected Huh7$_{A2}$HCV$_{EM}$ cells as analyzed by HLA-A2 BV650-specific flow cytometry. US, unstained control. Data are representative of $n = 3$ independent experiments. **c** Inhibition of viral replication was measured in IFN-expression vector-transfected compared to mock vector (MV) transfected Huh7$_{A2}$HCV$_{EM}$ cells by luciferase activity. Results are expressed as mean±SEM of $n = 4$ biological replicates from $n = 2$ independent experiments. $p < 0.0001$, MV vs IFNλ4; $p = 0.2487$, IFNλ1 vs IFNλ4; $p = 0.3448$, IFNλ3 vs IFNλ4, two-tailed unpaired $t$-test. **d** Peptide-mismatched Huh7$_{A2}$HCV cells were transfected exactly as described in (**a**). The transfected cells were then loaded with exogenous NS5B$_{2594-2602}$-ALYDVVTKL peptide before coculture with NS5B$_{2594-2602}$-specific CD8$^+$ T cells. Coculture and flow cytometry analysis were performed exactly as described in (**a**) except that the right panel shows the results of mean ± SEM from $n = 3$ independent experiments. IFNγ$^+$CD8$^+$: $p = 0.2810$, MV vs IFNλ4; TNF$^+$CD8$^+$: $p = 0.5873$, MV vs IFNλ4; CD107a$^+$CD8$^+$: $p = 0.5346$, MV vs IFNλ4, all by two-tailed unpaired $t$-test. Source data are provided as a Source Data file.

uninfected samples, mainly due to the induction of interferon-stimulated genes (ISGs). On the other hand, SeV infection-induced highly similar gene sets in both *dG* and *TT* organoids. Nevertheless, the upregulation of ER stress-related genes was stronger in organoids of the *dG* genotype, although the difference was not significant (Supplementary Fig. 8d). We conclude that the small amount of IFNλ4 expression in the context of viral infection does not significantly alter the overall transcriptional response induced by the very potent type I IFN system. However, this does not rule out that intracellular IFNλ4 retention causes ER stress and thereby affects protein trafficking.

The ER plays also an important role in the loading of viral antigen peptides onto the MHC class I complex. To test whether IFNλ4-induced ER stress could interfere with HCV antigen presentation, we used a unique in vitro cell coculture system consisting of sub-genomic HCV replicon cells stably transduced with HLA-A2 (Huh7$_{A2}$HCV$_{EM}$) and an HLA-matched HCV-specific CD8$^+$ T cell clone[26]. In this system, IFNλ4 expression in the antigen-presenting replicon cells impaired the CD8$^+$ T cell response. This was not due to inhibition of HCV replication or reduced cell surface expression of MHC class I molecules. Furthermore, we could rescue the CD8$^+$ T cell response by loading HCV peptide exogenously. This demonstrates that the negative effect of IFNλ4 on the CD8$^+$ T cell response is not mediated by direct inhibition of CD8$^+$ T cells, but by causing reduced antigen presentation. Moreover, when transfecting a plasmid expressing the non-functional IFNλ4-TT variant in the HCV replicon cells, CD8$^+$ T cell activity was not inhibited. Based on these observations, we propose that the IFNλ4 paradox can be explained by the negative impact of IFNλ4 on HCV antigen presentation to immune cells. Interestingly, previous studies have shown decreased numbers of CD8$^+$ T cells and CD163$^+$ macrophages in liver biopsies from patients of the IFNλ4 producing genotype[38], and decreased degranulation activity in freshly isolated CD3$^+$CD56$^-$ T and CD3$^+$CD56$^+$ NKT lymphocytes from liver biopsies of patients with an IFNλ4 producing genotype[39].

In conclusion, our study highlights a non-canonical effect of intracellularly retained IFNλ4. Small amounts of IFNλ4 are secreted, and these molecules are highly potent antiviral inducers of the JAK-STAT interferon signaling pathway. However, most of the IFNλ4 is retained in the ER and causes ER stress. ER stress interferes with HCV antigen peptide loading and surface presentation by MHC-I complexes. As a consequence, T cell responses are attenuated, favoring viral escape from the cellular immune response that is known to be crucial for viral clearance.

## Methods

**Reagent or resource.** Reagents used in this study are listed in Supplementary Table 1.

**Cell culture.** A549 (ATCC Number: CCL-185), Huh7, and Huh7-LR[16] cells were grown in low glucose (1 g/l) Dulbecco's modified Eagle (DMEM) medium (Gibco) supplemented with 10% fetal bovine serum (FBS) and 1% penicillin/streptomycin (10,000 U/ml) (ThermoFisher Scientific). Huh7.5[40] cells were grown in high glucose (4.5 g/l) Dulbecco's modified Eagle (DMEM) medium (Gibco) supplemented with 10% fetal bovine serum (FBS) and 1% penicillin/streptomycin. To generate IFNLR1-expressing Huh7.5.1 cells (Huh7.5.1-LR), Huh7.5.1 cells were transfected with pcDNA3.V5-IFNLR[16] before selection with 1 mg/ml G418. Tetracycline-inducible HepG2 cell lines stably expressing IFNA4-GFP or IFNA4 p131-GFP were gifts from Dr. Ludmila Prokunina-Olsson[41]. Cells were cultured in a DMEM medium containing 1 mg/ml G418 and 5 µg/ml blasticidin S hydrochloride. The replicon cells Huh7$_{A2}$HCV$_{EM}$, Huh7$_{A2}$HCV, and NS5B$_{2594-2602}$-specific CD8$^+$ T cell clone have previously been described[26]. Replicon cells were grown in high glucose (4.5 g/l) DMEM medium (Gibco) supplemented with 1% nonessential amino acids (Gibco), 1 mg/ml G418, and 3 µg/ml blasticidin S hydrochloride. The CD8$^+$ T cell clone was cultivated in RPMI 1640 medium (Gibco) supplemented with 10% human serum (PanBiotech), 1% L-glutamine, and stimulated twice a week with 30 U/ml interleukine-2 (IL-2). CD8$^+$ T cells were fed biweekly with irradiated peripheral blood mononuclear cells (PBMC) isolated from normal blood and stimulated with 40 µg/ml Phytohemagglutinin-M (PHA-M).

**Protein expression and purification.** Full-length open reading frames (ORF) for IFNα, IFNλ1, IFNλ2, IFNλ3, and IFNλ4 were amplified using the primers shown in Supplementary Table 2 and cloned into the pcDNA4/To/*myc*-His B expression vector (ThermoFisher Scientific) or into the pCMV-mir-GFP expression vector (a gift from Jacek Krol). The IFNλ4-TT expression plasmid was constructed by site-directed mutagenesis of "C$_{65}$" to "AA$_{65-66}$" in the IFNλ4 sequence. Recombinant IFN proteins were expressed in Huh7 cells by transient transfection with polyethylenimine (PEI) according to the manufacturer's instructions. Myc-His-tagged secreted IFNλ4, IFNλ3, and IFNα were purified from the culture supernatant 48 h post-transfection by Ni affinity chromatography using Ni Sepharose$^{TM}$ 6 fast gravity-flow resin (GE Healthcare) according to the manufacturer's instructions. Specifically, Myc-His-tagged proteins in supernatants were bound to the resin in buffer A (10 mM sodium phosphate, 10 mM sodium hydrogen phosphate, 500 mM NaCl, 5 mM imidazole, pH 7.8), and were eluted with buffer B (10 mM sodium phosphate, 10 mM sodium hydrogen phosphate, 500 mM NaCl, 500 mM imidazole, pH 7.4) after two times washing with buffers containing 10 mM and 20 mM imidazole, respectively. Purified proteins were concentrated with Amicon Ultra-15 filters (Merck Millipore) and dialyzed against phosphate-buffered saline (PBS). Because of the low amounts of IFNλ4 secreted from cells, 10-fold more supernatant was used for IFNλ4 than for IFNλ3 purification. The purified IFNλ4 and IFNλ3 preparations were finally further concentrated to 0.3 and 0.5 ml, respectively. The identity and purity of the purified proteins were analyzed by western blotting using anti-IFNλ4 (1:1000, Merck Millipore) and anti-IFNλ3 (1:1000, Abcam) antibodies (Supplementary Fig. 3a–c). Because of the much larger starting volume and final concentration necessary for IFNλ4 compared to IFNλ3 purification, the IFNλ4 preparation still contained some contaminating protein which we determined to be primarily carry over serum albumin that was present in the cell-culture supernatant used for protein purification (Supplementary Fig. 3a, b). Serum albumin however does not contribute to activation of the Jak-STAT interferon signaling pathway (Supplementary Fig. 3g). Because of the contaminating serum albumin, however, the IFNλ4 concentration in the purified preparation could not be directly measured. Therefore, the IFNλ4 concentration was determined by anti-Myc-specific (1:1000, Cell Signaling) western blotting using the purified IFNλ3 as a reference (Supplementary Fig. 3e). The concentration of purified IFNλ3 was determined independently with three methods: (i) Absorbance at 280 nm (Nanodrop) using a protein molar extinction coefficient of 11,500 and a molecular weight of 26 KDa, (ii) Bio-Rad protein assay (Bio-Rad) at absorbance 595 nm, (iii) 12% SDS-PAGE followed by silver staining using bovine serum albumin (BSA) (SigmaAldrich) as a standard (i.e., 50, 100, 200, and 400 ng) (Supplementary Fig. 3d). Each assay yielded very similar IFNλ3 concentrations and the one obtained

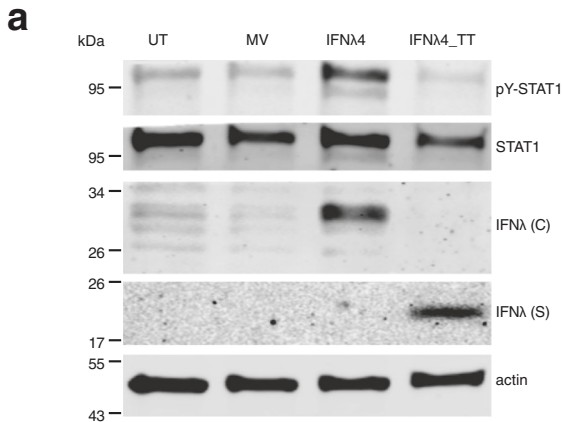

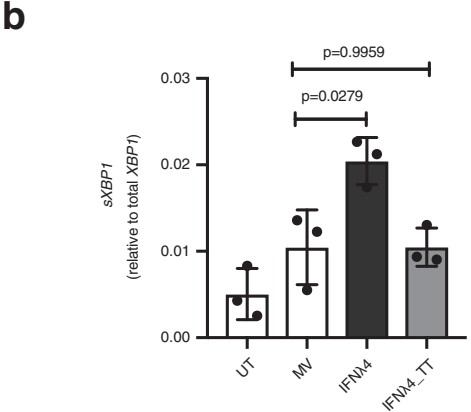

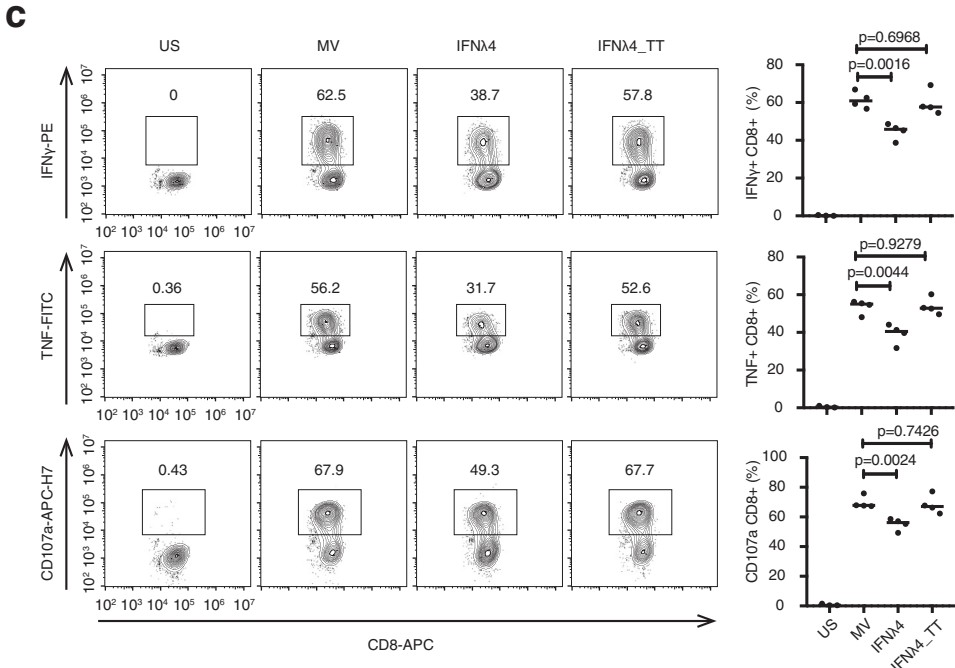

by silver staining was used for all subsequent experiments and quantification of purified IFNλ4. The quantification of the purified IFNλ4 and IFNλ3 preparation was confirmed by silver staining after separation by 12% SDS-PAGE (Supplementary Fig. 3f). E. coli-derived IFNλ3 and IFNλ4 proteins harboring N-terminal 6× His tags were kindly provided as a gift by Prof. Rune Hartmann[10]. The Fiji distribution of ImageJ[42] was used for protein quantification of silver staining and western blot images.

**Kinetics of expression and secretion of IFNλs.** IFNλ4 or IFNλ3 carrying PcDNA4/To/myc-His B plasmids were transiently transfected into Huh7 cells using PEI, in 5 wells of a 24-well plate, respectively. In the following 5 days, cells in one of the 5 wells were harvested every 24 h to monitor intracellular IFNλ accumulation. Cells were lysed in the culture plates in 200 μl Laemmli buffer containing β-mercaptoethanol (SigmaAldrich) and a protease inhibitor cocktail (PI, SigmaAldrich). In addition, supernatant (in total 250 μl) was collected and replaced with

**Fig. 7 IFNλ4-TT does not inhibit HCV peptide-specific CD8+ T cell activation. a** Huh7 cells were transfected with expression vectors encoding IFNλ4, IFNλ4-TT or a mock vector (MV) or left untreated (UT). Cell lysates and cell-culture supernatants were collected 24 h later. Cell lysates were analyzed by western blotting for STAT1 activation (pY-STAT1, STAT1), as well as intracellular IFNλ (IFNλ (C)) using an anti-Myc-tag antibody. IFNλ proteins in the supernatants (IFNλ (S)) were also detected with an anti-Myc-tag antibody. Data are representative of $n = 3$ independent experiments. **b** Huh7$_{A2}$HCV$_{EM}$ cells were transfected with expression vectors for IFNλ4 and IFNλ4-TT or a mock vector (MV) or were left untreated (UT). Total RNA was harvested 48 h after transfection. Expression of the ER stress marker sXBP1 was analyzed as described in Fig. 4d. Results are presented as mean ± SEM of $n = 3$ independent experiments. $p = 0.0279$, MV vs IFNλ4; $p = 0.9959$, MV vs IFNλ4_TT, two-tailed unpaired $t$-test. **c** Huh7$_{A2}$HCV$_{EM}$ cells were transfected as described in (**b**) and were 48 h later cocultured with the NS5B$_{2594-2602}$-specific CD8+ T cell clone for 5 h. Flow cytometry analysis was performed exactly as described in Fig. 6a. The right panel shows the results of mean ± SEM from $n = 4$ independent experiment. IFNγ+CD8+: $p = 0.0016$, MV vs IFNλ4, $p = 0.6968$, MV vs IFNλ4_TT; TNF+CD8+: $p = 0.0044$, MV vs IFNλ4, $p = 0.9279$, MV vs IFNλ4_TT; CD107a+CD8+: $p = 0.0024$, MV vs IFNλ4, $p = 0.7426$, MV vs IFNλ4_TT, all by two-tailed unpaired $t$-test. Source data are provided as a Source Data file.

fresh medium (250 µl) every 24 h until cells were harvested. Untransfected cells were used as a negative control. Proteins in 25 µl of each supernatant and 20 µl of each cell lysate were resolved by 10% SDS-PAGE and transferred onto a nitro-cellulose membrane (Waterman) by Trans-Blot Turbo Transfer System (Bio-Rad). Blotted proteins were detected with a monoclonal anti-Myc-tag antibody (1:1000, Cell Signaling) and visualized using a near-infrared-fluorescent dye (IRDye)-conjugated secondary anti-rabbit IgG (1:10,000, IRDye 800CW, Licor). Images were captured on an Odyssey-CLx image system (Licor). To visualize IFNλ4 in the supernatant, the same membranes were reblotted with a monoclonal anti-IFNλ4 antibody (1:1000, Merck Millipore) and a secondary anti-mouse IgG (1:10,000, IRDye 680RD, Licor). Protein signal intensities were quantified from the western blot images using Fiji.

**RNA extraction, reverse transcription, and quantitative real-time PCR**. Total RNA was extracted from cells using the Trizol reagent (Ambion) according to the manufacturer's instructions. Total RNA was subjected to DNase I treatment using the DNA-freeTM DNA Removal Kit (Ambion) following the manufacturer's instructions. RNA concentrations were determined using a NanoDrop 2000 spectrophotometer (ThermoFisher Scientific). cDNA was synthesized from 200 to 400 ng of total RNA using MultiScribe Reverse Transcriptase (Applied Biosystems) and random hexamer primers in a 25 µl reaction. Real-time quantitative PCR (RT-qPCR) was performed with 10% of each RT-reaction using the FastStart Universal SYBR Green Master (Roche Diagnostics) on an ABI 7500 or 7500 Fast Real-Time PCR System (Applied Biosystems). Primers used for RT-qPCR are listed in Supplementary Table 3. The amplification data were analyzed with the 7500 Software v2.0.6 (Applied Biosystems). Quantification of *IFNL4* gene expression was performed using the TaqMan Universal Master Mix (ThermoFisher Scientific). For a non-discriminative quantification of *IFNL4* transcripts, the TaqMan assay described in Amanzada et al.[9] was used. Forward and reverse primers are complementary to sequences with exon 3 (5′-GAGGGATGTGGCGGCCTG-3′) and exon 5 (5′-GACCACGCTGGCTTTGCG-3′), respectively, and a FAM-labeled minor groove binder (MGB) probe (5′-CCCGGAGAG CGGAC-3′) was designed to span the exon 4–5 boundary to further enhance PCR specificity. Target gene expression levels were quantified based on standard curves comprise of serial dilutions of plasmids containing either the corresponding complete cDNA or PCR amplicon region and expressed as copy numbers per 40 ng of total RNA. The lower limit of detection (LoD) for each RT-qPCR was set as the lowest detectable dilution of the corresponding standard curve. Alternatively, gene expression was expressed relative to that of the housekeeping gene *GAPDH*.

**JAK-STAT activity assay and ISG induction analysis**. For one-time point experiments, Huh7-LR cells were treated for 30 m or 6 h with either serial dilutions of purified IFNλ4 and IFNλ3 or specific amounts of IFNλ4 and IFNλ3 supernatants. For kinetic assays, Huh7-LR cells were treated with undiluted IFNλ4 supernatant and a 1:100 dilution of IFNλ3 supernatant for 0.5, 1, 2, 4, 8, 16, 24 h. For type I IFN inhibition with B18R, Huh7-LR cells were pre-incubated with B18R (R&D Systems,125, 250, 500, 1000 ng/ml) for 2 h. After washing, cells were then incubated with recombinant IFNα (Roferon-A) (250, 500, 1000 UI/ml) or with specific amounts of IFNλ4 and IFNλ3 supernatants for 30 m. Cells were lysed in Laemmli buffer and analyzed for total STAT1/2 protein, phospho-STAT1/2, and β-actin by western blotting. Antibodies were listed in the reagent table in Supplementary Table 1. Western blots were performed as described above. STAT activation was quantified as the ratio of phospho-STAT intensity to total STAT intensity using the Fiji application. Total RNA was isolated from cells treated for 6 h for ISG expression analysis by RT-qPCR as described above.

**IFN activity reporter assay**. Huh7-LR cells were electroporated using Cytomix[43,44] with 10 µg of the interferon-stimulated response element (ISRE)-Mx1 firefly luciferase reporter construct (pGL3-Mx1P-FF-Luc, a gift from Rune Hartmann). 18 h after electroporation, the cell-culture medium was replaced with fresh medium containing serial dilutions of IFNλ4 and IFNλ3. Cells were lysed 6 h later

in Passive Lysis Buffer (Promega) and firefly luciferase levels were measured using a multi-mode microplate reader (Centro XS3 LB960, Berthold Technologies).

**Hepatitis C virus infection**. Recombinant cell-culture-derived HCV (HCVcc) virus (strain JFH1/D183)[45] was generated as previously described[46]. Huh7.5.1-LR cells were infected with HCVcc for 6 h (MOI = 1) before adding IFNλ4 (1 and 10 ng/ml) or IFNλ3 (1, 10, and 50 ng/ml). Total cellular RNA was extracted 48 h post-infection and intracellular HCV RNA and ISG mRNA were analyzed by quantitative RT-qPCR.

**Protein deglycosylation**. Secreted and purified IFNλ4 was treated with PNGase F, O-glycosidase, β-N-acetyl glucosaminidase, β1–4 galactosidase or α2-3,6,8 neuraminidase individually, or with a protein deglycosylation mix (Mix-I, an enzyme mix of PNGase F, O-glycosidase, β-N-acetyl glucosaminidase, β1–4 galactosidase or α2-3,6,8 neuraminidase) or an enzyme mix of β-N-acetylglucosaminidase, β-1–4 galactosidase, neuraminidase (Mix-II) for 1 h at 37 °C. Cell lysates of IFNλ4-expressed Huh7 cells were produced in ice-cold sucrose homogenization medium (SHM) (sucrose 0.25 M, HEPES 10 mM, pH 7.4) containing a PI cocktail. Cell homogenates were then treated with PNGase F at 37 °C for 1 h. The corresponding molecular weights of the treated and control mock-treated IFNλ4 were visualized by western blotting with a monoclonal anti-IFNλ4 antibody (1:1000, Merck Millipore) as described above.

**Immunofluorescence microscopy**. Huh7 cells were cultured in 4-well chamber slides (Nunc Lab-Tek, SigmaAldrich). 24 h after transfection with pcDNA4/To/myc-His B-IFNλ4 plasmid or with pcDNA4/To/myc-His B vector, using PEI, cells were fixed with 4% paraformaldehyde and then permeabilized with 0.1% Triton X-100 in PBS containing 2% BSA. Fixed and permeabilized cells were incubated with anti-IFNλ4 (Merck Millipore, 5 µg/ml) and anti-Giantin antibodies (Abcam,1 µg/ml) in PBS containing 2% BSA for 1 h at 37 °C. After three times washing with PBS, the corresponding secondary antibodies (goat anti-rabbit 647, 1:1000, and goat anti-mouse 488, 1:400) were added under the same conditions as the primary antibodies. Excess antibodies were removed by three times washing with PBS. Endoplasmic reticulum (ER) staining was performed with the Cytopainter ER staining Kit-Red Fluorescence (Abcam) for 30 m at 37 °C. Slides were mounted with Roti-mount FlourCare DAPI (Carl Roth). Confocal microscopy was performed with a Zeiss LSM 710 confocal microscope and images were acquired with the ZEN 2010 software (Carl Zeiss). The colocalization coefficient (Pearson's coefficient) was quantified in ImageJ using the plug-in, JACoP (Just Another Colocalization Plug-in)[47].

**Subcellular fractionation**. The subcellular localization of IFNλ4 and IFNλ3 was determined in Huh7 cells 24 h after transfection with pcDNA4/To/myc-His B-IFNλ4 or with pcDNA4/To/myc-His B-IFNλ3, respectively, using PEI. $6 \times 10^7$ cells were resuspended in 2 ml ice-cold SHM containing a PI cocktail and homogenized in a Dounce homogenizer using a tight pestle. The cell homogenate (HM) was subjected to a 2-step fractionation procedure. First, differential centrifugation was used to sediment crude organelles components such as crude mitochondria (cMito), crude plasma membrane (cPM), microsome (MS), and cytosol (Cyto) fractions[48,49]. Second, the sedimented cMito, cPM, and MS fractions were resuspended in 0.5 ml Tris/EDTA buffer (TE) (10 mM TrisCl, 0.1 mM EDTA, PH 6.0) and subjected to density gradient centrifugation. Discontinuous iodixanol (OptiPrep, 60% wt/vol) gradients (5, 10, 15, 20, 30, 32.5, 35, and 40% from top to bottom) were prepared in a 5-ml ultra-clear SW55Ti centrifugation tube (Beckman Coulter)[50]. The resuspended cMito, cPM, and MS fractions were loaded on top of the iodixanol gradients, and centrifuged in an SW55Ti rotor at 200,000×g for 2 h. The centrifugation resulted in three visible bands. Fractions containing individual bands were collected and filled up to 12 ml with SHM/PI buffer, and pelleted by centrifugation in an SW32Ti rotor at 100,000 × g for 40 m. Pellets were resuspended in 250 µl SHM/PI. IFNλ4 and IFNλ3 in different compartments were quantified by Myc-tag-specific western blot analysis of 2 µl of each fraction (HM, PM, ER, ER-PM, Cyto) as described above. PM and ER markers were detected with

anti-E-cadherin (1:1000, Cell Signaling) and anti-calnexin (1:500, Santa-Cruz) antibodies, respectively. IFNλ4 and IFNλ3 protein concentrations in each fraction were quantified using purified IFNλ3 (25, 50, 100 ng) as a standard. STAT1/2 activation by the IFNλ4 and IFNλ3 containing fractions was analyzed as described above.

**ER stress analysis**. For western blot detection, Huh7 cells were transfected with IFN-expression plasmids using PEI as described above. Cells were harvested either at 24 h or for kinetic studies at 8, 12, 24, and 48 h after transfection and/or drug treatment and subjected to western blot analysis for the detection of IRE1α, phosphor- IRE1α, PERK, phosphor-PERK, Bip, cleaved ATF6 (cATF6), ATF4, phospho-eIF2α, eIF2α, and Myc-tagged IFNs. 4-15% Mini-PROTEAN TGX Stain-Free Precast Gels (Bio-Rad) and Trans-Blot Turbo Mini 0.2 μm Nitrocellulose Transfer Packs (Bio-Rad) were used for SDS-PAGE separation and Western blotting. For gene expression analysis, Huh7 cells were transfected with IFN-expression plasmids or incubated with IFN supernatants from transfected Huh7 cells at the indicated time points. Cellular RNA was extracted and used for RT-qPCR as described above. The expression level of spliced *sXBP1* mRNA was calculated relative to the expression level of total *XBP1* mRNA. *CHOP* mRNA expression was calculated relative to that of the housekeeping gene *GAPDH*. Huh-7 cells treated with 2 μM Thapsigargin (SigmaAldrich) for 4, 8, and 12 h served as a positive control for ER stress induction. HepG2 cell lines stably expressing IFNλ4-GFP or IFNλ4 p131-GFP were cultured in the presence of increasing amounts of doxycycline (0, 0.01, 0.02, 0.05, 0.1, 0.2, 0.5 μg/ml). After 24 h of doxycycline induction, IFNλ4-GFP and IFNλ4 p131-GFP expression in the cells and supernatant was quantified by western blotting using an anti-IFNλ4 antibody (1:1000, Merck Millipore) as described above.

**Organoid culture and hepatocyte differentiation**. Organoid cultures (Supplementary Table 4) were generated from liver-derived needle biopsies as previously described[24,51]. Briefly, liver-derived biopsies were digested with 2.5 mg/ml collagenase IV (Sigma), 0.1 mg/ml DNase (Sigma), and the released cells were seeded into droplets of reduced growth factor Basement Membrane Extract, type 2 (BME2) (Amsbio). After BME2 polymerization, expansion medium[51] was added to the cells. Liver organoids were passaged after dissociation with 0.25% Trypsin-EDTA (Gibco). For hepatocyte differentiation, liver organoids were kept for at least 5 days in an expansion medium supplemented with BMP7 (25 ng/ml) before switching to differentiation medium[51] for a period of 11–15 days with regular medium changes. Organoid lines are available upon request.

**DNA extraction and *IFNλ4* genotyping**. Total DNA was isolated from organoids using DNeasy Blood & Tissue Kit (Qiagen) according to the manufacturer's instructions. *IFNL4* genotyping was performed exactly as described previously[15]. A fragment of 850 base pair covering rs368234815 and rs117648444 was amplified with the Expand High Fidelity PCR System (Roche) using forward 5′-ACTGTGTGTGTGCTGTGCCTTC-3′ and reverse 5′-GGACGAGAGGGCCGTTA-GAG-3′ primers. The PCR product was purified using the NucleoSpin Gel and PCR clean-up kit (Macherey-Nagel AG) according to the manufacturer's instructions before sequencing (Microsynth AG).

**Sendai virus infection in cell lines and in liver-derived organoid**. A549 cells were infected with Sendai virus (H4 strain) (a gift from Prof Dominique Garcin of the University of Genève, Genève, Switzerland)[52] at MOI = 5. Cells were harvested at 3, 6, 9, 12, 24 h post-infection. Total cellular RNA was extracted with Trizol reagent and *IFNL4* mRNA expression was quantified by RT-PCR as described above. Expression analysis of endogenously induced IFNλ4 and IFNλ3 was analyzed by Western blotting as described above using cell lysates harvested 6 and 12 h post-infection.

Organoids were derived from liver biopsies of patients enrolled in this study. The study was carried out in accordance with The Code of Ethics of the World Medical Association (Declaration of Helsinki) and was approved by the Ethics Committee of North Western Switzerland (Authorization number EKNZ 2014–362). Written informed consent was obtained from all patients enrolled in this study. Organoids that underwent hepatocyte differentiation were removed from BME2 using cold PBS and washed 3 times with PBS. Suspended organoids were then infected with Sendai virus (H4 strain) at an MOI = 10 for 1 h before seeding them back into BME2 as described above. For RNA extraction, organoids were released from BME2 by incubating with 0.25% Trypsin for 5 min at 37 °C followed by Trypsin inactivation with DMEM/10%FBS, and total RNA was extracted using the ZR-Duet DNA/RNA miniprep kit (Zymo Research) according to the manufacturer's instructions. Sendai virus RNA and mRNAs of IFNs and ISGs were quantified by RT-qPCR as described above using the genes-specific primers shown in Supplementary Table 3. For the detection of endogenously produced IFNλ4 protein, organoids were released from BME2 by incubation in Cell Recovery Solution (Corning) and subsequent processing using the Mem-PER Membrane protein Extraction Kit (ThermoFisher Scientific) according to the manufacturer's instructions. The membrane fractions were used for IFNλ4-specific western blotting. The blots were first treated with Super Signal Western blot enhancer (ThermoFisher Scientific) for 10 m according to the manufacturer's instruction. After 5-times washing with double-distilled water (ddH₂O), the blots

were incubated with IFNλ4-specific antibody (1:1000, Abcam) and an HRP-conjugated goat anti-rabbit secondary antibody (1:5000, ThermoFisher Scientific). The image was captured on Gel Doc XR+ System (Bio-Rad).

**RNA sequencing and analysis**. RNA was extracted from six organoid lines (Supplementary Table 4) pre- and 12 h-post mock-or SeV infection as described above. 100 ng total RNA of each sample was used for RNA sequencing library preparation with NEBNext Ultra II Directional RNA Library Prep Kit for Illumina with sample purification beads (New England Biolabs) according to manufacturer's specifications. SR150 sequencing was performed on an Illumina NextSeq 550 System at the Institute of Medical Genetics and Pathology (University Hospital Basel) according to the manufacturer's guidelines. Primary data analysis was performed with the Illumina Real-Time Analysis software (RTA version 1.18.66.3). Sequence reads were aligned by STAR[53] using the two-pass approach to the human reference genome GRCh37. Gene quantification was performed using RSEM, a software package for estimating gene and iso-form expression levels from RNAseq data[54]. Gene expression analysis was performed using the edgeR R package[55] in RStudio v3.6. Specifically, genes with counts-per-million (cpm) < 1 in ≥3 samples were removed. Normalization was performed using the "TMM" (weighted trimmed mean) method. cpm values were log2-transformed prior to plotting by multidimensional scaling (MDS) using the top 500 pairwise variable genes. Differential gene expression analysis between SeV-infected and uninfected samples was performed using the quasi-likelihood *F*-test. Uninfected organoids at time points 0 and 12 h were treated as biological replicates. All tests were performed treating samples from the same organoids as paired samples. Genes with FDR ≤ 0.05 were considered differentially expressed. Representative analysis of Gene Ontology (GO) term was performed with the clusterProfiler package[56] in *dG* and *TT* organoids, respectively. Gene Set Enrichment analysis was conducted to compare differences between SeV-infected organoids of *dG* and *TT* genotypes in pathway levels, with the R-package "fgsea", version 1.14.0[57], and Gene Ontology-Biological Process (GO-BP) from MSigDB, version 7.2[58]. For fgsea, parameters nperm = 10,000 and minSize = 5 were set. As separating score, sign(logFC) * F-statistic was used, computed from the edgeR differential comparison of (dG.S-dG.PM) vs (TT.S-TT.PM). Leading genes of the selected pathway "GO_RESPONSE_TO_ENDOPLASMIC_RETICULUM_STRESS" were plotted as the log2 fold-change in expression levels induced by SeV infection in organoids of the *TT* genotype against the *dG* genotype.

**HCV replication and HLA-A2 quantification of HCV replicon cells**. $0.5 \times 10^5$ Huh7$_{A2}$HCV$_{EM}$ replicon cells/well were seeded in a 24-well plate one day before PEI transfection with a mock vector or expression plasmids for IFNλ1, IFNλ3, IFNλ4, and IFNλ4-TT or were incubated with IFNλ3 and IFNλ4 containing supernatants. Forty-eight hours post-transfection or incubation, HCV replication levels were determined by using the Steady-Glo Luciferase Assay System (Promega) according to the manufacturer's instructions. For surface HLA-A2 staining, cells were harvested 48 h post-transfection and stained with anti-human HLA-A2 BV650 antibody (1:50). Data was acquired using a CytoFLEX (Beckman) flow cytometer and analyzed with Flowjo 10.4 (FlowJo, LLC).

**HCV replicon cells and CD8$^+$ T cells coculture assay**. Huh7$_{A2}$HCV$_{EM}$ replicon cells were transfected or incubated with supernatant of IFNλ3 and IFNλ4 as described above. Forty-eight hours post-transfection, replicon cells were cocultured with NS5B$_{2594-2602}$-specific CD8$^+$ T cells at an effector-to-target ration (E/T) of 2:1 in a total of 500 μl medium in the presence of brefeldin A. After 5 h of coculture, cells were harvested. Live/dead cell staining was perform using a Zombi aqua fixable viability kit (Biolegend). After washing with 2% BSA, cells were stained with anti-CD8-APC (1:50) and anti-CD107a-APC-H7 (1:25) diluted in 2% BSA for 30 m at 4 °C. Thereafter, cells were fixed (IC fixation buffer, eBioscience) and permeabilized (permeabilization buffer, eBioscience) according to the manufacturer's instruction. Intracellular IFNγ and TNF staining was performed with anti-IFNγ-PE (1:20) and anti-TNFα-FITC (1:2) diluted in permeabilization buffer for 30 m at 4 °C. Huh7$_{A2}$HCV$_{EM}$ replicon cells were treated with 2 μM Thapsigargin for 24 h before coculturing with CD8$^+$ T cells as described above. Data was acquired using a CytoFLEX (Beckman) flow cytometer and analyzed with Flowjo 10.4. The cell gating strategy is shown in Supplementary Fig. 12. For T cell coculturing assay with the mismatched replicon cells, Huh7$_{A2}$HCV cells were transfected with the IFN-expression plasmids as described above. 48 h-post-transfection, the NS5B$_{2594-2602}$-peptide Alydvvtkl (10 μg/ml) was added to the cell culture medium for 1 h at 37 °C. Then, the cells were washed twice with a fresh medium before coculturing with the CD8$^+$ T cells in the presence of brefeldin A and subsequent flow cytometry analysis of the CD8$^+$ T cells as described above.

**Statistical analysis**. Data are presented as mean value ± SEM. Data were analyzed with Prism8 (GraphPad Software Inc, La Jolla, USA) using a student's two-tailed *t*-test for unpaired data. In all analyses, a two-tailed $p < 0.05$ (95% confidence interval) was considered statistically significant. Western blot, gel staining, immunofluorescence, and flow cytometry data are representative of three or more independent experiments.

**Reporting summary**. Further information on research design is available in the Nature Research Reporting Summary linked to this article.

## Data availability

All data that support the findings of this study are available within the manuscript and the supplementary information files or from the corresponding author upon reasonable request. The RNA-Seq data reported here have been deposited in the European Genome-Phenome Archive under primary accession number: EGAS00001005396. Source data are provided with this paper.

## Code availability

The computational code used for the RNAseq analysis is provided as Supplementary Methods in the Supplementary Information file.

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

## Acknowledgements

We thank Hans Henrik Gad and Rune Hartmann for providing recombinant human IFNλ4, Dominique Garcin for providing Sendai virus, and Ludmila Prokunina-Olsson for providing tetracycline-inducible HepG2 cell lines. This work was supported by Swiss National Science Foundation grants 310030B_185371 and 310030_166202 to M.H.H., by Deutsche Forschungsgemeinschaft grant TRR179, TP17 to V.L., and grant TR179, TP 1 to R.T.

## Author contributions

Q.C. designed and performed experiments, analyzed RNAseq data, wrote the manuscript. M.C.-L. designed and performed experiments of IFNλ4 expression, secretion, and JAK-STAT activation. A.S. designed and performed experiments of Sendai virus infection in liver-derived organoids. R.D.T. performed protein purification. I.F. and S.N. established liver-derived organoids. Designed and performed experiments of Sendai virus infection in liver-derived organoids. M.H., R.T., N.H., and V.L. provided HCV sub-genomic replicon cells and CD8+ T cell clones. C.K.Y.N. and G.R. analyzed RNAseq data. S.W. and M.H.H., designed and supervised the project and wrote the manuscript. M.H.H. obtained funding.

## Competing interests

The authors declare no competing interests.
