## [Peer Review File · Nature Communications]

REVIEWER COMMENTS

Reviewer #1 (Remarks to the Author):

Manuscript: Interferon lambda 4 induces endoplasmic reticulum stress, impairs viral antigen presentation and attenuates T cell responses

Comments: Qian Chen and co-workers investigated the biological mechanisms why some individuals with specific IL-28B genotype (IFNL4-dG) show impaired HCV clearance and treatment failure. The authors have extended the previous finding that this IFNL4-dG genotype expresses IFNL4 protein. They have presented several experimental evidence suggesting that IFNL4 protein is accumulated intracellularly and poorly secreted compared to IFNL3. The cell culture-derived IFNL4 protein seems more potent (show 28 fold) compared to IFNL3 inducing pStat1 and pStat2 activation. They found that intracellular IFNL4 is glycosylated. The glycosylated IFNL4 induces Stat phosphorylation better than non-glycosylated. These data presented in Figures 1-3 are convincing. Furthermore, the authors claim that the glycosylated IFNL4 retained in the Endoplasmic reticulum and cause ER stress. They show Western blot data for BIP, p-eIF2alpha, and mRNA level of sXBP, and CHOP. Weak sets of data presented to support this conclusion. The data claim that Sendai virus infection induces IFNL4 and ER-stress using liver organoids. There are no differences between virus replication and ISG induction between IFNL-dG and IFNL-TT genotype cells. The expression of lambda4 is not convincing. They show a similar band in the TT organoid culture. The impact of IFNL4 expression on global gene expression was examined and found no difference between IFNL4 genotypes. The final set of experiments shows IFNL4 induced ER stress could interfere with HCV antigen presentation; therefore reduce the activation of HCV-specific CD8+T cells. This data appears that IFNL4 expression reduced CD8+T cell activation by flow analysis. This study explains how IFNL4 speech impaired viral clearance at the level of attenuated T cell response. The manuscript is easy to follow and well written. Some of the data (shown in Figure 4-6) are weak, not convincing. The claim ER stress inhibiting CD8+ T cell activation, this mechanism is limited to IFN-lambda 4. HCV replication generates robust ER stress. They should whether this is related to IFNL4 expression.

No mechanistic studies have done to verify whether small molecule inhibitors targeting the ER stress or silencing L4 could reverse this HCV antigen loading and surface presentation of MHC. Additional data are supporting data needed to support these mechanisms. Please see my specific comments on the manuscript.

Specific comments:

Figure 1-3 data sets are apparent and convincing.

Figure 4 figure has many problems. Panel A needs quantification. Panel B is not consistent with IFNL4 present in ER since similar bands were present in lanes with IFNL3 expression.

Panel C is not convincing. The Western blot supporting the claim IFN-L4 expression induced ER stress needs dose-dependent analysis. The best way is to verify this question by analyzing the kinetic study, as they showed in Figure 1C. They should examine all three UPR axis why only BIP and XBP splicing were analyzed. Panel D results show differences are minimal.

Figure 5 panel B and C results are not convincing. Panel B, why IFNL4 is detected in IFNL4-TT cells (U16, nt5, B13). Organoids used in panel B and Panel C are not the same, and data are week. They should verify why the IFNL4-TT genotype produces IFNL4.

The claim that Sendai virus-induced IFNL4 induced XBP1 mRNA splicing need to verified by alternative method because the data shown in panel C is the week and not clear. Why was the HCV infection model not used?

Sendai virus is different from HCV. How they interpret the data of the Sendai virus in Figure 6? Since these data are not useful, it should go to the supplemental Figure.

Figure 7, panel A results are excellent. They have shown data of one point analysis. Can they show

some control data showing how ER stress impairs antigen presentation? Why these sets of the study were not performed using organoid cells with different IFNL4 genotypes. What is the IFNL4-genotype of replicon used?

The authors should provide a mechanistic explanation of whether ER stress, in general, impairs CD8+T cell response or Lambda specific ER stress does this. This part needs a more mechanistic study.

Is there any evidence showing that HCV infected people with IFN-dG genotype have impaired CD8+T cell response in general? IFNL4 specific mechanism should strengthen the observation.

Reviewer #2 (Remarks to the Author):

Review: Chen et al., – Interferon lambda 4 induces ER stress, impairs viral antigen presentation and attenuates T cell responses

Type III interferons (IFN lambdas) share the signaling pathway downstream of their specific receptor with the type I IFN family members, resulting in induction of an (almost) identical set of IFN stimulated genes and innate immune responses. Due to the tissue-restricted expression of the IFN lambda receptor on epithelial cells of the lungs, liver and GI tract, IFN lambda-mediated innate immune responses are of specific interest for studies addressing the antiviral immune response against viruses that have a tropism for such tissues, such as HCV, influenza virus, or norovirus. Previous GWAS have identified several SNPs in the IFN lambda locus that strongly associate with spontaneous and treatment-induced HCV clearance. An indel polymorphism in the IFNL4 gene results in a frame shift and thereby dictates whether IFN lambda 4 protein, which when recombinantly expressed has similar antiviral properties as other type III (or even type I) IFNs, can be made or not. However, a major conundrum in the field has been that the IFNL4 genotype allowing IFN lambda 4 protein expression associates with poor HCV clearance, while patients with the variant that cannot express IFN lambda 4 protein have a better prognosis; the so-called "IFN lambda 4 paradox".

In the presented study, Chen et al. set out to identify the molecular mechanism underlying the IFN lambda 4 paradox. They uncovered that cells able of expressing IFN lambda 4 protein show impaired MHC-I-mediated viral antigen presentation resulting in decreased CD8 T cell responses. This is caused by intracellular retention of IFN lambda 4 protein. More specifically, IFN lambda 4 protein is retained at the ER (potentially as misfolded protein) leading to UPR and ER stress, which in turn results in poor MHC-I processing at the ER.

The manuscript is well written and for the most part, the study is well designed, the experiments are sound and proper controls were included. The study confirms and expands on previous findings by others, and also adds important mechanistic data and findings explaining the IFN lambda 4 paradox. The use of primary liver organoids is a great addition to support the findings derived from cell lines and further supports the in vivo relevance. The following comments should be considered to further strengthen the authors interpretations and the manuscript:

Major:

1) ER stress can induce expression of IFNs and other inflammatory cytokines. These could have been carried over in the supernatant of IFN lambda 4 expressing cells. Ideally those pSTAT1 kinetics experiments should be done in IFNAR1 KO cells, or in presence of the pan-type I IFN inhibitor B18R.

2) Why is there a smear in the IFN lambda 4 protein purifications (e.g. Fig. 3A & S2A)? Similarly, the Ponceau staining in Fig. S2B shows a strong band around 70 kD in the purified protein. Why does it only occur in IFN lambda 4, but not lambda 3? How do the authors rule out that these impurities skew their results in the activity assays where IFN lambda 3 and 4 activities are

compared side-by-side (Fig. 2 & S3)?

3) I am not really convinced that the protein quantification in Fig. S2 is very accurate. The authors determine the concentration of recombinant IFN lambda 3 protein by silver stain against a BSA standard, and then use this IFN lambda 3 protein concentration to determine the concentration of recombinant IFN lambda 4, which is in my opinion very indirect and potentially inaccurate. A simple A280 measurement on a Nanodrop would be more accurate. I understand that the contaminants in the purified IFN lambda 4 protein might interfere with an absorbance measurement. Have the authors considered size exclusion for the recombinant protein?

4) The data on IFN lambda 4-mediated UPR in Figure 4 could be strengthened. The phosphorylation of eIF2a is only marginally different in IFN lambda 4 treated cells and I would expect a stronger phenotype. Overexpression of anything is known to induce UPR to some extent. Does the EV control express just the His/Myc-tag used in the other constructs or is it not expressing any coding sequence at all? It would be good to see a mock transfected control and ideally even a positive control for ER stress, such as thapsigargin, to show the actual magnitude of the observed changes. Alternatively, the WB membrane should be probed for other proteins indicative of UPR and ER-stress (e.g. XBP1 protein).

5) The data in Fig. 4C&D would benefit from the addition of samples treated with Golgi plug to test if the effect is specific to IFN lambda 4, or if for example IFN lambda 3 retained inside the cell would induce ER stress similarly.

6) A critical control for the proposed mechanism would be to include the IFN lambda 4 S70 variant for all experiments shown in Fig. 4A-D. GWAS have shown that S70 is beneficial for HCV clearance compared to the P70 wild type variant of IFNL4. Based on the author's hypothesis IFNL4 S70 expression should cause less ER stress, and potentially better MHC-I presentation. If so, is S70 retained at the ER to a lesser extent? Is it secreted better? Is the P70S genotype for the organoids known?

7) Why did the authors use Huh7 cells that overexpress the IFNLR1? Huh7 hepatoma cells express IFNLR1 and are very sensitive to its ligands.

Minor:

1) Fig. S5B: A positive control for IFN lambda 3 should also be included in the WB. This is especially important since the authors used a GelDoc system that (unlike film) automatically adjusts signal to noise ratio based on the strongest signal on the blot and doesn't truly represent cumulative signal intensity. In particular the blot for secreted IFN lambda 3 cannot be interpreted correctly in absence of the positive control, when in direct comparison the secreted IFN lambda 4 WB shows a positive control (which might mask the signal in the samples).

2) Why did the authors decide to not strip the Western blots in Fig. 1A & S1A? It makes it very difficult to appreciate which signal is actually coming from the anti-IFN lambda 4 antibody with the Myc/FLAG signal still bleeding through. It would have been better to first develop the IFN lambda 4, and then Myc/FLAG if stripping was to be absolutely avoided.

Reviewer #3 (Remarks to the Author):

It is a fascinating and thought-provoking study aimed to solve the "IFNL4 paradox" where the presence of biologically active antiviral cytokine is associated with worse outcomes for hepatitis C virus (HCV) infection in terms of low rates of spontaneous clearance and poor responsiveness to type I IFN-based therapies in patients chronically infected with HCV. Authors performed a set of diverse biochemical and immunological assays leading them to two major conclusions that native,

mammalian cell-produced IFN λ 4 is more biologically potent than other type III IFNs, and that the expression of IFN λ 4, which is purely secreted, results in the ER stress that causes impaired antigen presentation. The manuscript is well-written, logically presented and easy to follow, and the conclusions are important and highly imaginative. However the reviewer has several comments that needs to be addressed to substantiate the main conclusions.

Major:

1. Reviewer's main concern is that artificial overexpression systems are used to obtain results in most experiments, except experiments of Fig. 5. Although it is understandable that IFN-lambda4 studies are challenging, the authors seem to have a set of liver organoid cell lines of two different IFNL4 genotypes, either IFNL4-G/G (the functional IFNL4 gene) or IFNL4-TT/TT (the non-functional IFNL4 gene). These organoid cultures of the IFNL4-G/G genotype express endogenous IFN λ 4 at the physiological levels, unlike overexpression systems used in other experiments, at the levels which authors can detect after Sendai virus (SeV) infection. Although authors demonstrate that unfolded protein ER stress response is indeed detected in response to SeV infection only in organoids of the IFNL4-G/G genotype, the conclusions would be much stronger if authors could utilize these organoids for additional studies.

Specifically, these organoids cultures could be infected with HCV or at least transfected with HCV replicons to assess stress response in liver cells during HCV infection, the host cells and the virus where the IFNL4 genotype seems to play a particularly important role in virus-host interaction. Although the IFNL4-G/G genotype-dependent stress response was detected in organoids infected with SeV, it is important to confirm that this also happens during HCV infection. HCV may induce lower levels of IFN λ 4 expression that may not cause the ER stress.

Perhaps there are technical limitations that the reviewer may not be fully aware of for using this organoid system and HCV, but some additional experiments should be doable. For instance, experiments of Fig. 5A could be reproduced with organoids infected with HCV (transfected with HCV replicon), since SeV and HCV are likely to have different effects on timing and magnitude of IFN production and IFN response. In addition, authors should try to utilize organoids of different IFNL4 genotypes to confirm conclusions drawn from experiments of Fig. 7 to assess whether endogenous IFN λ 4 expressed at the physiological levels would have an effect on antigen presentation. Experiments of Fig. 7 use cells overexpressing not only IFN λ 4 but also HLA-A2. Thus, this system is highly artificial not only because of the overexpression of IFN λ 4, but also because HLA-A2 is also likely to be overexpressed or at least not affected by IFN λ production. Expression levels of endogenous MHC class I antigens are up-regulated by IFNs, but since HLA-A2 is expressed from a transfected construct, the IFN-induced upregulation of HLA-A2 was not detected in cells expressing IFN λ vs empty vector (Fig. 7B), although the Huh7 cells are responsive to IFN λ as can be judged from the suppression of the levels of HCV replicon (luciferase expression). Authors seem to have several available organoid cell lines (Fig. 5), perhaps some of them would have HLA-A2 and therefore could be potentially used for experiments with HLA-matched HCV-specific CD8+ T cell clone that was utilized in experiments of Fig. 7. Alternatively, antigen loading in organoids of different IFNL4 genotypes could be measured utilizing other immunological experimental tools that could be available for HCV or at least for other viral antigens. Also relevant, experiments of Fig. 7 should include cells not only transfected with IFN λ expression constructs but also cells treated with recombinant proteins as an additional control to rule out a possibility that the antigen load and/or T cell activation is altered at least in part by the secreted IFN λ 4 protein.

2. Authors confirm previous observations that IFN λ 4 is barely secreted from the cells and appear to be stuck in ER and Golgi apparatus. However the conclusion that mammalian cell-produced IFN λ 4 is more active than IFN λ 3 or E.coli cell-produced IFN λ 4 needs additional proof and clarification. Intracellular or secreted IFN λ 4 appears as two protein bands in most gels, as ~28kD band representing glycosylated form and ~20kD representing un-glycosylated form (for example Fig. 1A and S2C). However, when protein concentrations for mammalian cell-produced IFN λ 4 is normalized to that of IFN λ 3 only glycosylated form seems to be taken into account (Fig. S2E), whereas un-glycosylated IFN λ 4 is biologically active. More detailed description of how concentrations of purified proteins were determined needs to be provided to compare biological potencies of IFN λ 4 and IFN λ 3 proteins. Of note, although IFN λ 4 indeed seems to be more potent in terms of activation of STAT1 (levels of pSTAT1 induced by IFN λ 4 are stronger than those induced by IFN λ 3). However, these assays are done at one time point and differences in kinetics of STAT1 activation triggered by IFN λ 4 vs IFN λ 3 may partially account for this difference. Particularly questionable is an antiviral assay presented in Fig. S3C, where IFN λ 4 activity seems to plateau at

1ng/ml and did not increase at 10ng/ml (other concentrations were not assessed), whereas the antiviral activity of IFN λ 3 increases in a dose-dependent manner and at the concentration of 100ng/ml exceeds the antiviral activity of IFN λ 4 at concentrations used in this experiment. Therefore, it could be argued that at saturating concentrations IFN λ 3 has stronger antiviral activity than IFN λ 4. Of note results presented in Fig. 7C also appear to correlate with this observation since replication of HCV replicon in Huh7 cells overexpressing IFN λ 3 is suppressed to a higher extent than in cells overexpressing IFN λ 4.

Regarding activities of glycosylated vs non-glycosylated forms of IFN λ 4, a minor difference observed in experiments of Fig. 3D could be due to the buffer used to treat IFN λ 4 with PNGase. An additional control is required where the protein is incubated with the PNGase buffer without enzyme addition. Experiments comparing mammalian cell-produced and E.coli cell-produced IFN λ 4 and IFN λ 3 proteins need additional details and clarification. It is not explained why mammalian cell-produced IFN λ 3 has different MW than E.coli cell-produced IFN λ 3 (Fig. 3A). Both proteins are not glycosylated. Is it due to different protein tags or different N-terminus of the protein? Difference in MW of mammalian cell-produced IFN λ 4 and E.coli cell-produced IFN λ 4 are expected due to glycosylation that is only present in case of mammalian cell-produced form. Nevertheless, E.coli cell-produced IFN λ 4 has MW higher than non-glycosylated IFN λ 4 that is also present in mammalian cell-produced purified protein (Fig. S2C). Moreover, the sizes of E.coli cell-produced IFN λ 4 IFN λ 4 and IFN λ 3 appear identical in Fig. 3A, however their predicted MW are quite different. This needs to be explained and the identity of E.coli cell-produced IFN λ 4 confirmed with immunoblotting with IFN λ 4 specific Ab. In addition, comparing biological potencies of mammalian cell-produced and E.coli cell-produced IFN λ 4 may not be accurate since, as authors discuss elsewhere, differences in potencies may simply reflect the fact that a certain fraction of E.coli cell-produced IFN λ 4 may not properly folded, and the observed difference may not be due to the presence or the lack of protein glycosylation. Of note, experiments of Fig. 3B and 3D seem to correlate with results of experiments presented in Fig. 2. Glycosylated IFN λ 4 seems to be just slightly more active than non-glycosylated form of IFN λ 4 (~2-3-fold? Fig. 3B), which is as active as glycosylated or non-glycosylated forms of IFN λ 3 (Fig. 3D). Then how results presented in Fig. 2 indicate that glycosylated IFN λ 4 was approximately 28-fold more potent than glycosylated IFN λ 3. Minor

1. The use of p131 form of IFN λ 4 needs to be explained in detail. Reference to the publication of Prokunina-Olsson et al. (Fig. 2 in this previous report) indicates that this form seems to be lacking exon 3 and somehow two amino acids at the end of exon 2 (unusual alternative splicing?). This form is biologically inactive. Curiously, authors' data indicates that this form also accumulates intracellularly, likely in ER, as the mature functional IFN λ 4 does (Fig. S4B). However, this p131 form of IFN λ 4 does not trigger ER stress response (Fig. S4B). Therefore, it could be argued that the activation of the stress response requires a biological activity of IFN λ 4, and not due to the fact that the protein appears to be stuck in ER since both forms are located intracellularly. This needs to be further explored. Is the intracellular location of the p131 form of IFN λ 4 IFN λ 4 is different than the full length IFN λ 4 IFN λ 4 protein? What is unique within the missing part of the protein encoded by the exon 3 that may trigger stress response?

2. Of note, experiments of Fig. 4D indicate that "expression of CHOP, a key transcription factor that is important to initiate the apoptotic program in response to excessive ER stress, was not significantly induced by IFN λ 4 expression (Fig. 4D)" (lines 200-2002). However, experiments of Fig. S4B demonstrate the opposite. Please comment and conciliate.

3. It is somewhat puzzling that although stress response is observed only in liver cell organoids with the IFNL4-G/G genotype (Fig. 5C), it does not translate into a unique transcriptional response (Fig. 6). The reviewer would expect that a set of stress response genes would be induced only in organoids of the IFNL4-G/G genotype and not of the IFNL4-TT/TT genotype. Can authors specifically focus on the transcriptional profiles of genes induced by the stress response?

4. Line 151: It is stated that IFN λ 3 is physiologically not glycosylated. That's correct for N-linked glycosylation. The presence of O-linked glycosylation remains to be investigated and is likely to be present as authors data indicate based on different mobility of cytoplasmic and secreted IFN λ 3 forms (Fig. 1A).

We thank the reviewers for their insightful comments and suggestions that helped us to improve this manuscript. Below please find our point-by-point responses. Page numbers refer to the revised manuscript text file with highlighted changes.

REVIEWER COMMENTS

Reviewer #1 (Remarks to the Author):

Manuscript: Interferon lambda 4 induces endoplasmic reticulum stress, impairs viral antigen presentation and attenuates T cell responses

Comments: Qian Chen and co-workers investigated the biological mechanisms why some individuals with specific IL-28B genotype (IFNL4-dG) show impaired HCV clearance and treatment failure. The authors have extended the previous finding that this IFNL4-dG genotype expresses IFNL4 protein. They have presented several experimental evidence suggesting that IFNL4 protein is accumulated intracellularly and poorly secreted compared to IFNL3. The cell culture-derived IFNL4 protein seems more potent (show 28 fold) compared to IFNL3 inducing pStat1 and pStat2 activation. They found that intracellular IFNL4 is glycosylated. The glycosylated IFNL4 induces Stat phosphorylation better than non-glycosylated. These data presented in Figures 1-3 are convincing. Furthermore, the authors claim that the glycosylated IFNL4 retained in the Endoplasmic reticulum and cause ER stress. They show Western blot data for BIP, p-eIF2alpha, and mRNA level of sXBP, and CHOP. Weak sets of data presented to support this conclusion. The data claim that Sendai virus infection induces IFNL4 and ER-stress using liver organoids. There are no differences between virus replication and ISG induction between IFNL-dG and IFNL-TT genotype cells. The expression of lambda4 is not convincing. They show a similar band in the TT organoid culture. The impact of IFNL4 expression on global gene expression was examined and found no difference between IFNL4 genotypes. The final set of experiments shows IFNL4 induced ER stress could interfere with HCV antigen presentation; therefore reduce the activation of HCV-specific CD8+T cells. This data appears that IFNL4 expression reduced CD8+T cell activation by flow analysis. This study explains how IFNL4 speech impaired viral clearance at the level of attenuated T cell response. The manuscript is easy to follow and well written. Some of the data (shown in Figure 4-6) are weak, not convincing. The claim ER stress inhibiting CD8+ T cell activation, this mechanism is limited to IFN-lambda 4. HCV replication generates robust ER stress. They should whether this is related to IFNL4 expression. No mechanistic studies have done to verify whether small molecule inhibitors targeting the ER stress or silencing L4 could reverse this HCV antigen loading and surface presentation of MHC. Additional data are supporting data needed to support these mechanisms. Please see my specific comments on the manuscript.

Specific comments:

Figure 1-3 data sets are apparent and convincing.

Reviewer 1; Comment 1: We are pleased that the reviewer liked these figures and the presented data. We would like to inform the reviewer that figures 1 and 2 nevertheless have been slightly modified in response to comments of the other reviewers.

Figure 4 figure has many problems. Panel A needs quantification.

Reviewer 1; Comment 2: As suggested, we quantified co-localization of IFN λ 4 with ER and with golgi using JACoP, a Plug-in of Fiji. The analysis confirmed the preferential accumulation of IFN λ 4 in the ER. The new data is described in the text (page 8) and in the revised figure 4a.

Panel B is not consistent with IFN λ 4 present in ER since similar bands were present in lanes with IFN λ 3 expression.

Reviewer 1; Comment 3: We are not sure that we understand this comment. IFN λ 4 is almost exclusively detected in the ER and ER-PM fractions, whereas IFN λ 3 is mainly found in the cytosolic fraction. We agree that a small amount of IFN λ 3 is also found in the ER-PM fraction, but this does not change our conclusion that IFN λ 4 is retained in the ER.

Panel C is not convincing. The Western blot supporting the claim IFN- λ 4 expression induced ER stress needs dose-dependent analysis. The best way is to verify this question by analyzing the kinetic study, as they showed in Figure 1C. They should examine all three UPR axis why only BIP and XBP splicing were analyzed. Panel D results show differences are minimal.

Reviewer 1; Comment 4: We thank the reviewer for this comment. As suggested, we performed kinetic studies (0, 8h, 24h, 48h) after transfection of IFN λ 3 or IFN λ 4 in Huh7 cells, and examined all three UPR axis by western blotting, including p-IRE1 α , IRE1 α , p-PERK, PERK, Bip, ATF6, ATF4, p-eIF2 α , eIF2 α . PERK, IRE1, ATF6 and ATF4 were not significantly activated upon expression of IFN λ 4 in Huh7 cells. Of note, treatment of Huh7 cells with the ER stress inducer Thapsigargin did also not affect these proteins. On the contrary, there was a transient induction of eIF2 α with a peak at time point 24h in the IFN λ 4 transfected cells. We also examined mRNA expression of sXBP1 and CHOP in kinetic studies. Huh-7 cells treated with 2 μ M Thapsigargin for 4, 8 and 12 hours served as a positive control for ER stress induction. We found a IFN λ 4 specific induction of sXBP1, but no differences in CHOP induction. To address the question of the magnitude of ER stress induction, we compared IFN λ 4 induced sXBP1 amounts with Thapsigargin induced sXBP1. We found comparable induction levels and we conclude that the difference between IFN λ 4 and IFN λ 3 is not minimal. The new data is described in the text (page 9) and shown in the revised figure 4c-d (and in supplementary figure 5c, previous panel D). Previous panels C and D of the original Figure 4 were moved to the new Supplementary figure 5b-c.

Figure 5 panel B and C results are not convincing. Panel B, why IFN λ 4 is detected in IFN λ 4-TT cells (U16, nt5, B13). Organoids used in panel B and Panel C are not the same, and data are weak. They should verify why the IFN λ 4-TT genotype produces IFN λ 4.

Reviewer 1; Comment 5: We think that there might be a misunderstanding. There is no IFNL4 band in IFNL4-TT organoids (U16 and nt5). The lower band that appears in all lanes, including all uninfected samples, is a nonspecific signal. Only the upper band (arrow pointed) is the IFNL4 protein.

Panel b and c: The results shown in a and b were obtained in a first experiment with 4 organoid lines, 2 dG and 2 TT. To test for ER stress induction, additional organoid lines were used in a next experiment (panel c).

Panel c shows the amounts of sXBP1 and IFNL4 mRNA as circles connected with lines and as bars, respectively. IFNL4 mRNA (bars) is induced in both dG and TT organoids to similar extents. (In dG organoids, the IFNL4 mRNA codes for the functional IFNL4 protein, in the TT organoids the IFNL4 mRNA codes for the truncated, non-functional IFNL4-TT protein). On the contrary, splicing of XBP1 occurs only in dG organoids (red circles, top panel) but not in TT organoids (black circles, bottom panel).

The claim that Sendai virus-induced IFNL4 induced XBP1 mRNA splicing need to verified by alternative method because the data shown in panel C is the week and not clear. Why was the HCV infection model not used?

Reviewer 1; Comment 6: We hope that we answered the first sentence of the comment with our answer to the previous comment.

We did not use the HCV infection model because organoids cannot be infected with HCV (unfortunately; please see our more extensive comments on this topic in our general remarks at the beginning of this point-to-point reply).

Sendai virus is different from HCV. How they interpret the data of the Sendai virus in Figure 6? Since these data are not useful, it should go to the supplemental Figure.

Reviewer 1; Comment 7: As suggested we have moved original figure 6 to the supplementary figures (with modifications suggested by other reviewers). Sendai Virus and HCV activate the interferon system through the same sensory pathway, the RIG-I – MAVS - IRF-3/IRF-7 pathway. Contrary to HCV that can infect a very restricted number of cells in culture (all of them derived from the hepatoma cell line Huh7), Sendai virus can infect virtually any cell. Therefore, Sendai virus has been used widely to study innate immune responses to RNA viruses. As outlined above, liver organoids cannot be infected with HCV, but they can be infected with Sendai virus. The experiments shown in the old figure 6 show that despite the induction of IFNL4 by Sendai virus in dG organoids (as shown in figure 5b), the global transcriptional response to Sendai virus is very similar to that observed in TT organoids. However, ER-stress-related pathway was positively enriched in organoids of dG genotype, although the difference did not reach statistical significance. Nevertheless, the regulation of the corresponding genes was stronger in the organoids of the dG genotype. We discuss the conclusions and implications of this “negative” finding in the manuscript (page 10-12). We agree that the data can be shown as a supplemental figure (new Supplementary figure 8).

Figure 7, panel A results are excellent. They have shown data of one point analysis. Can they show some control data showing how ER stress impairs antigen presentation? Why these

sets of the study were not performed using organoid cells with different IFNL4 genotypes. What is the IFNL4-genotype of replicon used?

Reviewer 1; Comment 8: We are pleased that the reviewer liked the figure and the presented data.

We have done additional experiments using an ER stress inducer (Thapsigargin). As expected, Thapsigargin induced XBP1 splicing and indeed, HCV specific CD8+ T cell activation was significantly reduced. The new data is described in the text (page 11) and in the new Supplementary figure 9.

We fully agree that the organoid experiment would be the optimal approach. However, organoids cannot be infected with HCV and do not support replication of HCV replicons. Since this is an important issue, we would like to explain it in more detail:

After the original description of human liver organoids (Huch et al., 2015), we immediately set out to establish liver organoids from human liver biopsies in our lab with the principle aim to infect these organoids with hepatitis C and hepatitis B viruses. In the following years, huge efforts were undertaken by several postdocs and PhD students in the lab. Unfortunately, we had to realize that human liver organoids are not useful for HCV or HBV infection studies. Below is a short list of these efforts and failures:

- Over the years, we infected 24 different liver organoid lines with cell culture derived HCV (that can infect Huh7 cells), but not a single one could be productively infected.*
- We checked for HCV entry receptors: they are expressed in liver organoids.*
- We hypothesized that insufficient expression of miR122 (an important host factor that supports HCV replication in hepatocytes) might prevent productive HCV replication, and therefore stably over-expressed miR122 in organoids.*
- We incubated organoids with HCV positive patient sera.*
- We transfected HCV replicons into liver organoids: they did not replicate.*
- We also generated liver organoids from liver biopsies of patients with chronic hepatitis C: We observed some (carry-over) HCV RNA in the primary cultures, but HCV RNA was not detectable any more after the first passage of the cultures.*
- We performed a series of analogous experiments with HBV, again without success.*

We know from other research groups that they also failed to infect liver organoids with hepatitis viruses. There are also no reports in the scientific literature that demonstrate feasibility of infecting liver organoids with HCV or HBV.

Given our negative experience in the past and the lack of any encouraging reports from the scientific community, we cannot confirm the Sendai Virus liver organoid experiments with HCV infection or HCV replicon studies in liver organoids.

To circumvent the fact that organoids with different genotypes cannot be used to study the HCV specific T cell response, we used the Huh7 replicon cell system in

additional experiments. We transfected the replicon cells not only with the IFN λ 4-dG but also with the IFN λ 4-TT expression construct. This allowed a direct comparison of the impact of IFN λ 4 genotype on T cell responses. Cells transfected with an IFN λ 4-TT expression plasmid showed no activation of the Jak-STAT pathway, no ER stress response and did not inhibit HCV specific CD8+ T cell responses. These new data are described in the revised text (page 12) and in the new figure 7 and Supplementary figure 11.

Huh7 replicon cells have the IFN λ 4-TT genotype.

The authors should provide a mechanistic explanation of whether ER stress, in general, impairs CD8+T cell response or Lambda specific ER stress does this. This part needs a more mechanistic study.

Reviewer 1; Comment 9: We hope that we answered the comment with our answer to the previous comment. We have done additional experiments using an ER stress inducer (Thapsigargin). As expected, Thapsigargin induced XBP1 splicing and indeed, HCV specific CD8+ T cell activation was significantly reduced. The new data is described in the text (page 11) and in the new Supplementary figure 9.

Is there any evidence showing that HCV infected people with IFN-dG genotype have impaired CD8+T cell response in general? IFNL4 specific mechanism should strengthen the observation.

Reviewer 1; Comment 10: We thank the reviewer for the comment. This is a very interesting question. Indeed, work that we cite as references 32 and discuss in the second last paragraph of the discussion (page 15) has shown decreased numbers of CD8+ T cells in livers of IFNL4-dG genotypes. And more importantly, work that we cite as reference 33 has shown decreased degranulation activity in fresh isolated CD3+ T cells from liver biopsies of IFNL4-dG patients.

Reviewer #2 (Remarks to the Author):

Review: Chen et al., – Interferon lambda 4 induces ER stress, impairs viral antigen presentation and attenuates T cell responses

Type III interferons (IFN lambdas) share the signaling pathway downstream of their specific receptor with the type I IFN family members, resulting in induction of an (almost) identical set of IFN stimulated genes and innate immune responses. Due to the tissue-restricted expression of the IFN lambda receptor on epithelial cells of the lungs, liver and GI tract, IFN lambda-mediated innate immune responses are of specific interest for studies addressing the antiviral immune response against viruses that have a tropism for such tissues, such as HCV, influenza virus, or norovirus. Previous GWAS have identified several SNPs in the IFN lambda locus that strongly associate with spontaneous and treatment-induced HCV clearance. An indel polymorphism in the IFNL4 gene results in a frame shift and thereby dictates whether IFN lambda 4 protein, which when recombinantly expressed has similar antiviral properties as other type III (or even type I) IFNs, can be made or not. However, a major conundrum in the field has been that the IFNL4 genotype allowing IFN lambda 4 protein expression associates with poor HCV clearance, while patients with the variant that cannot express IFN lambda 4 protein have a better prognosis; the so-called “IFN lambda 4 paradox”.

In the presented study, Chen et al. set out to identify the molecular mechanism underlying the IFN lambda 4 paradox. They uncovered that cells able of expressing IFN lambda 4 protein show impaired MHC-I-mediated viral antigen presentation resulting in decreased CD8 T cell responses. This is caused by intracellular retention of IFN lambda 4 protein. More specifically, IFN lambda 4 protein is retained at the ER (potentially as misfolded protein) leading to UPR and ER stress, which in turn results in poor MHC-I processing at the ER. The manuscript is well written and for the most part, the study is well designed, the experiments are sound and proper controls were included. The study confirms and expands on previous findings by others, and also adds important mechanistic data and findings explaining the IFN lambda 4 paradox. The use of primary liver organoids is a great addition to support the findings derived from cell lines and further supports the in vivo relevance. The following comments should be considered to further strengthen the authors interpretations and the manuscript:

Major:

1) ER stress can induce expression of IFNs and other inflammatory cytokines. These could have been carried over in the supernatant of IFN lambda 4 expressing cells. Ideally those pSTAT1 kinetics experiments should be done in IFNAR1 KO cells, or in presence of the pan-type I IFN inhibitor B18R.

Reviewer 2; Comment 1: We thank the reviewer for this important comment. We have repeated the experiment to assess the activity of the secreted IFNλ4 in presence of the pan-type I IFN inhibitor B18R as suggested (new figure 2a-c and Supplementary figure 2). Importantly, treatment of Huh7-LR cells with B18R

prevented IFN α - but not IFN λ 3- and IFN λ 4-mediated JAK-STAT activation. This data is described on page 6 of the revised manuscript.

2) Why is there a smear in the IFN lambda 4 protein purifications (e.g. Fig. 3A & S2A)? Similarly, the Ponceau staining in Fig. S2B shows a strong band around 70 kD in the purified protein. Why does it only occur in IFN lambda 4, but not lambda 3? How do the authors rule out that these impurities skew their results in the activity assays where IFN lambda 3 and 4 activities are compared side-by-side (Fig. 2 & S3)?

Reviewer 2; Comment 2: The reviewer correctly points out that the purified IFN λ 4 preparation contains impurities that are not present in the IFN λ 3 preparation. Due to the intracellular retention of IFN λ 4 (shown in Fig. 1), the supernatant of IFN λ 4 transfected Huh7 cells contains at least 100-fold less IFN λ 4 than IFN λ 3 as shown in Fig. 2a. We therefore had to start with a large volume of supernatant that contained minute amounts of IFN λ 4. As a consequence, contaminating proteins could not be separated sufficiently during the purification procedure (a problem well recognized in the field (Pubmed 18235434)). In our case, the major contaminating protein of 70 kD is in fact serum albumin carried over from the cell culture medium. This is now shown in the revised supplementary Fig. 3a and b. To formally rule out that albumin activates the Jak-STAT pathway, we performed the additional experiment shown in revised supplementary Fig. 3g.

The complete purification procedure is now more clearly described in the Materials and Methods section of the revised manuscript (page 19-20) and shown in the revised supplementary figure 3. As explained in the response to comment 3 below, we also tried alternative purification procedures, but could not improve the purity of the IFN λ 4 preparation.

Of note, since the discovery of IFN λ 4 in 2013, to our knowledge, our report is the first one to describe the production and usage of purified secreted IFN λ 4 obtained from a mammalian system.

3) I am not really convinced that the protein quantification in Fig. S2 is very accurate. The authors determine the concentration of recombinant IFN lambda 3 protein by silver stain against a BSA standard, and then use this IFN lambda 3 protein concentration to determine the concentration of recombinant IFN lambda 4, which is in my opinion very indirect and potentially inaccurate. A simple A280 measurement on a Nanodrop would be more accurate. I understand that the contaminants in the purified IFN lambda 4 protein might interfere with an absorbance measurement. Have the authors considered size exclusion for the recombinant protein?

Reviewer 2; Comment 3: We agree with the reviewer that it would be optimal to directly determine the protein concentration of both the purified IFN λ 4 and IFN λ 3. However, as the reviewer noted, the contaminating proteins in the IFN λ 4 preparation don't allow a useful direct measurement. As we have explained in the response to comment 2 above, the very low concentration of IFN λ 4 in the input supernatant inevitably increases the co-purification of proteins unspecifically bound to the resin.

Nevertheless, in collaboration with Dr. Raphael Dias Teixeira, a protein structure biologist at the Biocentre Basel, we have pursued alternative purification approaches. However, all approaches failed to yield a purer IFN λ 4 preparation. As suggested by the reviewer, we also included a size exclusion step in the various purification strategies. But despite its small molecular weight, IFN λ 4 always co-eluted with much larger proteins (e.g. serum albumin). We suspect that, under the required purification conditions, IFN λ 4 might form a multimer itself or is complexed to other proteins (maybe the serum albumin) and thus evades efficient separation from the other proteins.

We agree with the reviewer that the stepwise quantification of IFN λ 3 against BSA and then using IFN λ 3 as the reference for IFN λ 4 quantification is rather indirect. We would like to point out, however, that the main purpose of the purification of IFN λ 3 and IFN λ 4 was to compare their relative activities. Thus, our primary interest was to ensure that we use comparable amounts of IFN λ 3 and IFN λ 4. We therefore quantified them by Western blotting with an anti-myc antibody (supplementary fig. 3e).

4) The data on IFN lambda 4-mediated UPR in Figure 4 could be strengthened. The phosphorylation of eIF2 α is only marginally different in IFN lambda 4 treated cells and I would expect a stronger phenotype. Overexpression of anything is known to induce UPR to some extent. Does the EV control express just the His/Myc-tag used in the other constructs or is it not expressing any coding sequence at all? It would be good to see a mock transfected control and ideally even a positive control for ER stress, such as thapsigargin, to show the actual magnitude of the observed changes. Alternatively, the WB membrane should be probed for other proteins indicative of UPR and ER-stress (e.g. XBP1 protein).

Reviewer 2; Comment 4: As suggested we have performed a number of additional experiments to strengthen the data. Specifically, we included Thapsigargin as a positive control for ER stress and as negative controls we added untreated samples and samples transfected with a mock vector (not expressing any coding sequence). We also expanded the panel of ER stress markers. Furthermore, we monitored all ER stress markers in a kinetic experiment over 48 hours. As described in the response to comment 4 of reviewer 1, PERK, IRE1, ATF6 and ATF4 were not significantly activated upon expression of IFN λ 4 in Huh7 cells. Of note, treatment of Huh7 cells with the ER stress inducer Thapsigargin did also not affect these proteins. In contrast, there was a transient induction of eIF2 α with a peak at time point 24h in the IFN λ 4 transfected cells.

We also examined mRNA expression of sXBP1 and CHOP in kinetic studies. Huh-7 cells treated with 2 μ M Thapsigargin for 4, 8 and 12 hours served as a positive control for ER stress induction. We found an IFN λ 4 specific induction of sXBP1 to a similar level as induced by Thapsigargin.

The new data is described in the text (page 9) and shown in the revised figure 4c-d (and in supplementary figure 5c, previous panel D).

Previous panels C and D of the original Figure 4 were moved to the new Supplementary figure 5b-c.

5) The data in Fig. 4C&D would benefit from the addition of samples treated with Golgi plug

to test if the effect is specific to IFN lambda 4, or if for example IFN lambda 3 retained inside the cell would induce ER stress similarly.

*Reviewer 2; Comment 5: We thank the reviewer to raise this interesting question. As suggested, we repeated the ER stress induction experiments in the presence of brefeldin A (golgi plug). Huh7 cells were left untreated (UT) or transfected with an empty mock vector or an IFNλ3 or IFNλ4 expression vector. 16 hours after transfection, the cells were treated with brefeldin A (BFA) (Biolegend, 5 μg/ml) or left untreated. The cells and supernatants were harvested 8 hours later and protein extracts and total RNA were used ER stress and IFNλ3/4 specific Western blotting and sXBP1 mRNA analysis, respectively. As shown in Figure 1 below, BFA prevented secretion of IFNλ3 (Sec IFN) leading to intracellular IFNλ3 accumulation (Intra IFN) while IFNλ4 accumulated intracellularly independent of the BFA treatment. Again, as described in the revised manuscript, the strongest ER stress response was observed for sXBP1 mRNA (Figure * included in the point-to-point reply, right panel). However, BFA treatment alone induced a strong induction of sXBP1 mRNA to levels that were as high as those observed in the IFNλ3 and IFNλ4 expressing cells treated with BFA. Thus, it remains unclear to what extent intracellular retention of IFNλ3 causes ER stress. Therefore, we did not include this data in the revised manuscript.*

Figure *

6) A critical control for the proposed mechanism would be to include the IFN lambda 4 S70 variant for all experiments shown in Fig. 4A-D. GWAS have shown that S70 is beneficial for HCV clearance compared to the P70 wild type variant of IFNL4. Based on the author's hypothesis IFNL4 S70 expression should cause less ER stress, and potentially better MHC-I presentation. If so, is S70 retained at the ER to a lesser extent? Is it secreted better? Is the P70S genotype for the organoids known?

Reviewer 2; Comment 6: We agree with the reviewer that this is an interesting point. We have performed these experiments previously. In the overexpression system (Huh7 cells transfected with expression plasmids for IFNλ3, IFNλ4-P70 and IFNλ4-

S70), there was no consistent difference between P70 and S70. In both cases, we observed equally strong ER retention. We then hypothesized that differences between P70 and S70 would be revealed in the more physiological organoid system. One of the organoids, nt115, indeed is homozygous for S70 (Supplementary Table 3). However, as shown in Figure 5c, there is considerable variation of IFNL4 mRNA expression between organoid lines. The degree of XBP1 splicing seems to correlate more with the amount of IFNL4 mRNA than with the P70 versus S70 genotype. Thus, we concluded that differences between P70 and S70 are too subtle to be detected in our in vitro experimental systems.

7) Why did the authors use Huh7 cells that overexpress the IFNLR1? Huh7 hepatoma cells express IFNLR1 and are very sensitive to its ligands.

Reviewer 2; Comment 7: There are considerable differences between the Huh7 cell lines used in different laboratories. Our Huh7 cells respond very poorly to IFN λ , and we therefore generated a series of Huh7 derived cell lines constitutively expressing IFNLR1 (Reference 15: Duong, Trincucci et al, JEM, 2014), and used them in the present work as IFNL reporter cells.

Minor:

1) Fig. S5B: A positive control for IFN lambda 3 should also be included in the WB. This is especially important since the authors used a GelDoc system that (unlike film) automatically adjusts signal to noise ratio based on the strongest signal on the blot and doesn't truly represent cumulative signal intensity. In particular the blot for secreted IFN lambda 3 cannot be interpreted correctly in absence of the positive control, when in direct comparison the secreted IFN lambda 4 WB shows a positive control (which might mask the signal in the samples).

Reviewer 2; Comment 8: We very much agree with the reviewer. We repeated the experiment with a proper IFN λ 3 control. The result is shown in revised Supplementary figure 6b.

2) Why did the authors decide to not strip the Western blots in Fig. 1A & S1A? It makes it very difficult to appreciate which signal is actually coming from the anti-IFN lambda 4 antibody with the Myc/FLAG signal still bleeding through. It would have been better to first develop the IFN lambda 4, and then Myc/FLAG if stripping was to be absolutely avoided.

Reviewer 2; Comment 9: We apologize for the unclear presentation of the Western blot data in Figure 1a. Binding of the secondary antibodies was performed at the same time with antibodies coupled to different fluorophores. The anti-Myc and anti-IFN λ 4 signals were then detected in the proper fluorescence channels. In the original Figure we showed only the Myc signal in the upper panel and both signals (Myc and IFN λ 4) in the bottom panel. In the revised Figure 1a, we now only show the IFN λ 4 signal in the bottom panel.

Reviewer #3 (Remarks to the Author):

It is a fascinating and thought-provoking study aimed to solve the “IFN λ 4 paradox” where the presence of biologically active antiviral cytokine is associated with worse outcomes for hepatitis C virus (HCV) infection in terms of low rates of spontaneous clearance and poor responsiveness to type I IFN-based therapies in patients chronically infected with HCV. Authors performed a set of diverse biochemical and immunological assays leading them to two major conclusions that native, mammalian cell-produced IFN λ 4 is more biologically potent than other type III IFNs, and that the expression of IFN λ 4, which is purely secreted, results in the ER stress that causes impaired antigen presentation. The manuscript is well-written, logically presented and easy to follow, and the conclusions are important and highly imaginative. However the reviewer has several comments that needs to be addressed to substantiate the main conclusions.

Major:

1. Reviewer’s main concern is that artificial overexpression systems are used to obtain results in most experiments, except experiments of Fig. 5. Although it is understandable that IFN-lambda4 studies are challenging, the authors seem to have a set of liver organoid cell lines of two different IFNL4 genotypes, either IFNL4-G/G (the functional IFNL4 gene) or IFNL4-TT/TT (the non-functional IFNL4 gene). These organoid cultures of the IFNL4-G/G genotype express endogenous IFN λ 4 at the physiological levels, unlike overexpression systems used in other experiments, at the levels which authors can detect after Sendai virus (SeV) infection. Although authors demonstrate that unfolded protein ER stress response is indeed detected in response to SeV infection only in organoids of the IFNL4-G/G genotype, the conclusions would be much stronger if authors could utilize these organoids for additional studies.

Specifically, these organoids cultures could be infected with HCV or at least transfected with HCV replicons to assess stress response in liver cells during HCV infection, the host cells and the virus where the IFNL4 genotype seems to play a particularly important role in virus-host interaction. Although the IFNL4-G/G genotype-dependent stress response was detected in organoids infected with SeV, it is important to confirm that this also happens during HCV infection. HCV may induce lower levels of IFN λ 4 expression that may not cause the ER stress.

Perhaps there are technical limitations that the reviewer may not be fully aware of for using this organoid system and HCV, but some additional experiments should be doable. For instance, experiments of Fig. 5A could be reproduced with organoids infected with HCV (transfected with HCV replicon), since SeV and HCV are likely to have different effects on timing and magnitude of IFN production and IFN response. In addition, authors should try to utilize organoids of different IFNL4 genotypes to confirm conclusions drawn from experiments of Fig. 7 to assess whether endogenous IFN λ 4 expressed at the physiological levels would have an effect on antigen presentation. Experiments of Fig. 7 use cells overexpressing not only IFN λ 4 but also HLA-A2. Thus, this system is highly artificial not only because of the overexpression of IFN λ 4, but also because HLA-A2 is also likely to be overexpressed or at least not affected by IFN λ production. Expression levels of endogenous MHC class I antigens are up-regulated by IFNs, but since HLA-A2 is expressed from a transfected construct, the IFN-induced upregulation of HLA-A2 was not detected in cells expressing IFN λ vs empty vector (Fig. 7B), although the Huh7 cells are responsive to IFN λ as can be judged from the suppression of the levels of HCV replicon (luciferase expression).

Authors seem to have several available organoid cell lines (Fig. 5), perhaps some of them would have HLA-A2 and therefore could be potentially used for experiments with HLA-matched HCV-specific CD8+ T cell clone that was utilized in experiments of Fig. 7. Alternatively, antigen loading in organoids of different IFNL4 genotypes could be measured utilizing other immunological experimental tools that could be available for HCV or at least for other viral antigens.

Reviewer 3; Comment 1.1: We fully agree with this reviewer and reviewer 1 that the organoid system would be ideal for these studies. Unfortunately, organoids cannot be infected with HCV and do not support HCV replicon replication. Since this is an important issue, we would like to explain it in more detail:

After the original description of human liver organoids (Huch et al., 2015), we immediately set out to establish liver organoids from human liver biopsies in our lab with the principle aim to infect these organoids with hepatitis C and hepatitis B viruses. In the following years, huge efforts were undertaken by several postdocs and PhD students in the lab. Unfortunately, we had to realize that human liver organoids are not useful for HCV or HBV infection studies. Below is a short list of these efforts and failures:

- Over the years, we infected 24 different liver organoid lines with cell culture derived HCV (that can infect Huh7 cells), but not a single one could be productively infected.*
- We checked for HCV entry receptors: they are expressed in liver organoids.*
- We hypothesized that insufficient expression of miR122 (an important host factor that supports HCV replication in hepatocytes) might prevent productive HCV replication, and therefore stably over-expressed miR122 in organoids.*
- We incubated organoids with HCV positive patient sera.*
- We transfected HCV replicons into liver organoids: they did not replicate.*
- We also generated liver organoids from liver biopsies of patients with chronic hepatitis C: We observed some (carry-over) HCV RNA in the primary cultures, but HCV RNA was not detectable any more after the first passage of the cultures.*
- We performed a series of analogous experiments with HBV, again without success.*

We know from other research groups that they also failed to infect liver organoids with hepatitis viruses. There are also no reports in the scientific literature that demonstrate feasibility of infecting liver organoids with HCV or HBV.

Given our negative experience in the past and the lack of any encouraging reports from the scientific community, we cannot confirm the Sendai Virus liver organoid experiments with HCV infection or HCV replicon studies in liver organoids.

To circumvent the fact that organoids with different genotypes cannot be used to study the HCV specific T cell response, we used the Huh7 replicon cell system in additional experiments. We transfected the replicon cells not only with the IFNL4-dG but also with the IFNL4-TT expression construct. This allowed a direct comparison of the impact of IFNL4 genotype on T cell responses. Cells transfected with an IFNL4-

TT expression plasmid showed no activation of the Jak-STAT pathway, no ER stress response and did not inhibit HCV specific CD8+ T cell responses. These new data are described in the revised text (page 12) and in the new figure 7 and Supplementary figure 11.

Also relevant, experiments of Fig. 7 should include cells not only transfected with IFN λ expression constructs but also cells treated with recombinant proteins as an additional control to rule out a possibility that the antigen load and/or T cell activation is altered at least in part by the secreted IFN λ 4 protein.

Reviewer 3; Comment 1.2: As suggested by the reviewer, we have done additional control experiments showing that treatment with secreted IFN λ 3 or IFN λ 4 did not induce an ER stress response in the HCV replicon cells and had no impact on T cell activation. We include these data in the text of the revised manuscript (page 11) and in the new supplementary figure 9.

2. Authors confirm previous observations that IFN λ 4 is barely secreted from the cells and appear to be stuck in ER and Golgi apparatus. However the conclusion that mammalian cell-produced IFN λ 4 is more active than IFN λ 3 or E.coli cell-produced IFN λ 4 needs additional proof and clarification. Intracellular or secreted IFN λ 4 appears as two protein bands in most gels, as ~28kD band representing glycosylated form and ~20kD representing unglycosylated form (for example Fig. 1A and S2C). However, when protein concentrations for mammalian cell-produced IFN λ 4 is normalized to that of IFN λ 3 only glycosylated form seems to be taken into account (Fig. S2E), whereas unglycosylated IFN λ 4 is biologically active.

*Reviewer 3; Comment 2.1: We thank the reviewer for the comment. However, this comment might be based on a misunderstanding. First, it has been previously demonstrated that unglycosylated IFN λ 4 is not secreted (Ref 10) and our results in Fig. 3c confirm this. Second, the lower bands in the purified IFN λ 4 lane in the original Supplementary Figure 2C are smaller than 20kD and represent IFN λ 4 degradation products. To better clarify this, we repeated the IFN λ 4-specific Western blot with a more detailed protein marker clearly showing that the lower bands in purified IFN λ 4 lane (PP) are smaller than 20kD (new Supplementary Figure 3c and Figure ** included in the point-to-point reply). For comparison, we also included an IFN λ 4 containing cell extract (C) showing the apparent molecular weight of the unglycosylated (~24kD) and partially glycosylated (~27kD) IFN λ 4. Therefore, bands smaller than 20 kD should not be taken into account for IFN λ 4 quantification.*

*Figure ***

More detailed description of how concentrations of purified proteins were determined needs to be provided to compare biological potencies of IFNλ4 and IFNλ3 proteins.

Reviewer 3; Comment 2.2: As requested, we provide a more detailed description of IFNλ3 and IFNλ4 purification and quantification in the revised "Materials and Methods" (page 19-20).

Of note, although IFNλ4 indeed seems to be more potent in terms of activation of STAT1 (levels of pSTAT1 induced by IFNλ4 are stronger than those induced by IFNλ3). However, these assays are done at one time point and differences in kinetics of STAT1 activation triggered by IFNλ4 vs IFNλ3 may partially account for this difference.

Reviewer 3; Comment 2.3: We agree with the reviewer and performed a kinetics of STAT1/2 activation by IFNλ3 and IFNλ4. These results confirm that IFNλ4 is more potent in inducing STAT1 and STAT2 phosphorylation compared to IFNλ3. This data is described in the revised manuscript (page 6) and shown in figure 2d-e.

Particularly questionable is an antiviral assay presented in Fig. S3C, where IFNλ4 activity seems to plateau at 1ng/ml and did not increase at 10ng/ml (other concentrations were not assessed), whereas the antiviral activity of IFNλ3 increases in a dose-dependent manner and at the concentration of 100ng/ml exceeds the antiviral activity of IFNλ4 at concentrations used in this experiment. Therefore, it could be argued that at saturating concentrations IFNλ3 has stronger antiviral activity than IFNλ4.

Reviewer 3; Comment 2.4: We thank the reviewer for the comment. However, we think that this is a misunderstanding. IFNλ3 was used at a maximal dose of 50 ng/ml. The 100 IU/ml belongs to IFNα. The main point of the figure (which is now moved to supplementary figure 4e in the revised manuscript) is that IFNλ4 at 1 ng/ml is as effective as 50 ng/ml of IFNλ3 in terms of antiviral activity.

Of note results presented in Fig. 7C also appear to correlate with this observation since replication of HCV replicon in Huh7 cells overexpressing IFN λ 3 is suppressed to a higher extent than in cells overexpressing IFN λ 4.

Reviewer 3; Comment 2.5: We respectfully disagree on this point. Although the replication in the IFN λ 4 transfected cells is slightly higher compared to IFN λ 1 and IFN λ 3 transfected cells, the difference is minor and not significant. We include p-values in the revised figure 6c to clarify this.

Regarding activities of glycosylated vs non-glycosylated forms of IFN λ 4, a minor difference observed in experiments of Fig. 3D could be due to the buffer used to treat IFN λ 4 with PNGase. An additional control is required where the protein is incubated with the PNGase buffer without enzyme addition.

Reviewer 3; Comment 2.6: We apologize for not having clearly stated the meaning of 'mock treated'. Indeed, the mock treatment included GlycoBuffer without any enzymes. We have included this information in the revised figure legend.

Experiments comparing mammalian cell-produced and E.coli cell-produced IFN λ 4 and IFN λ 3 proteins need additional details and clarification. It is not explained why mammalian cell-produced IFN λ 3 has different MW than E.coli cell-produced IFN λ 3 (Fig. 3A). Both proteins are not glycosylated. Is it due to different protein tags or different N-terminus of the protein?

Reviewer 3; Comment 2.7: We thank the reviewer for the comment. The differences are due to different tags. We have clarified this now in the revised figure legend (figure 3a) and in the "Materials and Methods" of the revised text (page 20).

Difference in MW of mammalian cell-produced IFN λ 4 and E.coli cell-produced IFN λ 4 are expected due to glycosylation that is only present in case of mammalian cell-produced form. Nevertheless, E.coli cell-produced IFN λ 4 has MW higher than non-glycosylated IFN λ 4 that is also present in mammalian cell-produced purified protein (Fig. S2C).

*Reviewer 3; Comment 2.8: We think this comment might be based on a misunderstanding. As we explained in the response to comment 2.1 above, non-glycosylated IFN λ 4 is not secreted and the molecular weights of the lower bands in the purified IFN λ 4 (PP) lane are smaller than non-glycosylated IFN λ 4 present in the cellular extract (Supplementary Figure 3c and Figure ** included in the point-to-point reply) and represent IFN λ 4 degradation products.*

*The expression vectors for IFN λ 4 production in E. coli and mammalian cells include a N-terminal His tag and C-terminal Myc/His tag, respectively. Thus, the apparent molecular weight of E. coli produced IFN λ 4 is smaller than that of the intracellular unglycosylated IFN λ 4 form in Huh7 cells (Supplementary Figure 3c and Figure ** included in the point-to-point reply).*

Moreover, the sizes of E.coli cell-produced IFN λ 4 and IFN λ 3 appear identical in Fig. 3A, however their predicted MW are quite different. This needs to be explained and the

identity of E.coli cell-produced IFN λ 4 confirmed with immunoblotting with IFN λ 4 specific Ab.

*Reviewer 3; Comment 2.9: The predicted MW of IFN λ 4 and IFN λ 3 are very similar with 19 kDa and 21 kDa, respectively. Both recombinant proteins are larger due to the N-terminal 6x His tag. The resolution of the SDS PAGE shown in Figure 3a is too low to resolve the small size difference between the two proteins. As requested, we confirmed the identity of E. coli produced IFN λ 4 by IFN λ 4-specific Western blotting (Figure ** included in the point-to-point reply).*

In addition, comparing biological potencies of mammalian cell-produced and E.coli cell-produced IFN λ 4 may not be accurate since, as authors discuss elsewhere, differences in potencies may simply reflect the fact that a certain fraction of E.coli cell-produced IFN λ 4 may not properly folded, and the observed difference may not be due to the presence or the lack of protein glycosylation.

Reviewer 3; Comment 2.10: We thank the reviewer for the comment. We think that it is very unlikely that E.coli produced IFN λ 4 is not properly folded, because it is equally active as IFN λ 3 produced in E.coli and mammalian cells.

Of note, experiments of Fig. 3B and 3D seem to correlate with results of experiments presented in Fig. 2. Glycosylated IFN λ 4 seems to be just slightly more active than non-glycosylated form of IFN λ 4 (~2-3-fold? Fig. 3B), which is as active as glycosylated or non-glycosylated forms of IFN λ 3 (Fig. 3D). Then how results presented in Fig. 2 indicate that glycosylated IFN λ 4 was approximately 28-fold more potent than glycosylated IFN λ 3.

Reviewer 3; Comment 2.11: Simple quantification of the signals in figure 3b revealed in fact a 5 fold difference between E.coli and Huh7 produced IFN λ 4 (and the two IFN λ 3 samples (data not shown)). To more accurately compare the potency of IFN λ 3 and IFN λ 4 , we did dose response curve experiments (previously shown in Fig 2, no shown in Supplementary Figure 4). Based on these experiments, we calculated an EC50 for both proteins, and the difference between them was 28-fold.

Minor

1. The use of p131 form of IFN λ 4 needs to be explained in detail. Reference to the publication of Prokunina-Olsson et al. (Fig. 2 in this previous report) indicates that this form seems to be lacking exon 3 and somehow two amino acids at the end of exon 2 (unusual alternative splicing?). This form is biologically inactive. Curiously, authors' data indicates that this form also accumulates intracellularly, likely in ER, as the mature functional IFN λ 4 does (Fig. S4B). However, this p131 form of IFN λ 4 does not trigger ER stress response (Fig. S4B). Therefore, it could be argued that the activation of the stress response requires a biological activity of IFN λ 4, and not due to the fact that the protein appears to be stuck in ER since both forms are located intracellularly. This needs to be further explored. Is the intracellular location of the p131 form of IFN λ 4 IFN λ 4 is different than the full length IFN λ 4 IFN λ 4 protein? What is unique within the missing part of the protein encoded by the exon 3 that may trigger stress response?

Reviewer 3; Comment 3: We used the tet-inducible HepG2 cells from Ludmilla Prokunina-Olsson to verify induction of ER stress by IFN λ 4 in an additional experimental system. The p131 expressing cells were used as a control. As outlined by the reviewer, p131 was also retained inside the cell. Compared to IFN λ 4, the amount of intracellular p131 was lower. We can speculate that the lack of sXBP1 induction is a consequence of this quantitative difference. We agree that a further characterization of p131 is of interest, but we think it is beyond the scope of the current study.

2. Of note, experiments of Fig. 4D indicate that “expression of CHOP, a key transcription factor that is important to initiate the apoptotic program in response to excessive ER stress, was not significantly induced by IFN λ 4 expression (Fig. 4D)” (lines 200-2002). However, experiments of Fig. S4B demonstrate the opposite. Please comment and conciliate.

Reviewer 3; Comment 4: We think that the differences might come from using different cells (Huh7 versus HepG2).

3. It is somewhat puzzling that although stress response is observed only in liver cell organoids with the IFN λ 4-G/G genotype (Fig. 5C), it does not translate into a unique transcriptional response (Fig. 6). The reviewer would expect that a set of stress response genes would be induced only in organoids of the IFN λ 4-G/G genotype and not of the IFN λ 4-TT/TT genotype. Can authors specifically focus on the transcriptional profiles of genes induced by the stress response?

Reviewer 3; Comment 5: We thank the reviewer for this suggestion. For the revised manuscript, we performed a gene set enrichment analysis on the genes induced by SeV infection in the IFN λ 4-dG and IFN λ 4-TT organoids. This analysis revealed that the pathway ‘GO-Response_To_Endoplasmic_Reticulums_Stress’ was positively enriched in organoids of dG genotype, although the difference did not reach statistical significance. Nevertheless, the regulation of the corresponding genes was stronger in the organoids of the dG genotype consistent with the induction of an IFN λ 4 dependent ER stress upon viral infection. We included the new data in the text (page 10-11) and in the new Supplementary figure 8d.

4. Line 151: It is stated that IFN λ 3 is physiologically not glycosylated. That’s correct for N-linked glycosylation. The presence of O-linked glycosylation remains to be investigated and is likely to be present as authors data indicate based on different mobility of cytoplasmic and secreted IFN λ 3 forms (Fig. 1A).

Reviewer 3; Comment 6: We agree that studying O-linked glycosylation of IFN λ 3 is interesting, but we believe that it is outside of the scope of the current study.

REVIEWERS' COMMENTS

Reviewer #1 (Remarks to the Author):

This is a difficult area of study. We need to know more about the Interferon lambda 4 expression contributes on innate and adaptive immunity. This study at least propose one mechanism how IFN-lambda contributes to impaired T cell response. The authors have addressed my concerns. I accept the paper.

Reviewer #2 (Remarks to the Author):

The authors have carried out several experiments that sufficiently address this reviewer's concerns. They have done a great job addressing the comments during COVID which is commendable. While the ER stress phenotype is mild, the authors show a strong antigen presentation phenotype. Before publication, this reviewer expects textual changes (and not experimental evidence) while addressing the comments below.

The data presented on ER stress is mostly based on the overexpression data is not striking. However this does not rule out ER stress, but the authors should acknowledge that in their discussion.

This study shows IFNL primary activity is a non-canonical one. This should be clearly stated in their discussion. While this reviewer appreciates the use of the term "paradox of IFNL4", there are multiple SNPs in high LD in this region that might affect viral clearance. Authors should acknowledge other genetic studies on IFNL genes in this loci to tone down a single gene hypothesis in their discussion. Several genes in innate immunity are either induced weakly or are pseudogenes. IFNL4 is not the first such described gene in innate immunity that is regulated in this way. Please rewrite parts of the discussion to include these themes.

The first 3 figures are confirmatory data that have been previously published from multiple studies. This reviewer proposes that the authors should integrate these three figures into one that only shows the major/new observations. The rest of the panels in figures 1-3 should be moved into the supplemental data. The authors should also rigorously cite previous work which this study has elegantly confirmed. Doing this will allow the authors to highlight their new observations of ER stress and antigen presentation.

Reviewer #3 (Remarks to the Author):

Authors have performed additional experiments and analysis suggested by referees, provided extended description for experimental procedures, and also clarified most points raised in the referee's comments. In my opinion the revised version of the manuscript is now suitable for publication in Nature Communications.

REVIEWERS' COMMENTS

Reviewer #1 (Remarks to the Author):

This is a difficult area of study. We need to know more about the Interferon lambda 4 expression contributes on innate and adaptive immunity. This study at least propose one mechanism how IFN-lambda contributes to impaired T cell response. The authors have addressed my concerns. I accept the paper.

Answer: We thank the reviewer for his positive response to our revised manuscript.

Reviewer #2 (Remarks to the Author):

The authors have carried out several experiments that sufficiently address this reviewer's concerns. They have done a great job addressing the comments during COVID which is commendable. While the ER stress phenotype is mild, the authors show a strong antigen presentation phenotype. Before publication, this reviewer expects textual changes (and not experimental evidence) while addressing the comments below.

The data presented on ER stress is mostly based on the overexpression data is not striking. However this does not rule out ER stress, but the authors should acknowledge that in their discussion.

Answer: We agree with the reviewer that many of the ER stress response markers are not very strongly induced in the Huh-7 cells. However, these markers, with the exception of CHOP, were not more strongly induced by the potent ER stress inducer Thapsigargin (Fig. 4c and 4d). We therefore think that the weak induction of these ER stress response markers rather reflects an intrinsic property of the Huh-7 cells than a limited ER stress inducing activity of IFN λ 4, because the IFN λ 4 mediated induction of sXBP1 mRNA was as strong as that triggered by the potent ER stress inducer Thapsigargin (Fig. 4d). We now state this in the discussion (page 13).

This study shows IFNL primary activity is a non-canonical one. This should be clearly stated in their discussion. While this reviewer appreciates the use of the term "paradox of IFNL4", there are multiple SNPs in high LD in this region that might affect viral clearance. Authors should acknowledge other genetic studies on IFNL genes in this loci to tone down a single gene hypothesis in their discussion. Several genes in innate immunity are either induced weakly or are pseudogenes. IFNL4 is not the first such described gene in innate immunity that is regulated in this way. Please rewrite parts of the discussion to include these themes.

Answer: We agree that the IFNL4 activity is non-canonical, and mention this now explicitly in the discussion (page 14). We also acknowledge in the discussion that some of the other SNPs with high LD in the region could be functional and could influence the host response to HCV (page 12).

The first 3 figures are confirmatory data that have been previously published from multiple studies. This reviewer proposes that the authors should integrate these three figures into

one that only shows the major/new observations. The rest of the panels in figures 1-3 should be moved into the supplemental data. The authors should also rigorously cite previous work which this study has elegantly confirmed. Doing this will allow the authors to highlight their new observations of ER stress and antigen presentation.

Answer: While we agree with the reviewer that part of the data confirms previous findings, we believe that our work adds substantial new information. For example, we demonstrate that IFN λ 4 does not accumulate extracellularly even after 5 days of expression (Figure 1) while previous studies (Hamming et al., 2013, Lu et al., 2015a, Hong et al., 2016) have only demonstrated poor secretion of IFN λ 4 at a single time point. Figure 2 summarizes the data of the quantitative comparison of the relative activity of secreted mammalian IFN λ 4 and IFN λ 3, both produced in the same Huh7 transfection system. This has not been done before. Finally, Figure 3 and Supplementary Figure 3 show the successful production and purification of secreted authentic mammalian recombinant IFN λ 4 in Huh7 cells. Thus, in contrast to previous studies (Hamming et al., 2013, Hong et al., 2016, Obajemu et al., 2017) that relied on recombinant IFN λ 4 produced in non-mammalian systems (e.g. *E. coli* and *Drosophila* S2 Schneider cells), we could show that the authentic IFN λ 4 is much more active than IFN λ 3 and that this is at least in part due to its glycosylation (Figure 3 and Supplementary Fig. 4). For these reasons we believe that the figures are warranted as currently presented.

Reviewer #3 (Remarks to the Author):

Authors have performed additional experiments and analysis suggested by referees, provided extended description for experimental procedures, and also clarified most points raised in the referee's comments. In my opinion the revised version of the manuscript is now suitable for publication in Nature Communications.

Answer: We thank the reviewer for supporting the publication of our revised manuscript in Nature Communications.